# A Bias-Variance-Covariance Decomposition of Kernel Scores for Generative Models

## Abstract

Generative models, like large language models, are becoming increasingly relevant in our daily lives, yet a theoretical framework to assess their generalization behavior and uncertainty does not exist. Particularly, the problem of uncertainty estimation is commonly solved in an ad-hoc manner and task dependent. For example, natural language approaches cannot be transferred to image generation. In this paper we introduce the first bias-variance-covariance decomposition for kernel scores and their associated entropy. We propose unbiased and consistent estimators for each quantity which only require generated samples but not the underlying model itself. As an application, we offer a generalization evaluation of diffusion models and discover evidence of how mode collapse is a contrary phenomenon to overfitting. Further, we demonstrate that variance and predictive kernel entropy are viable measures of uncertainty for image, audio, and language generation. Specifically, our approach for uncertainty estimation is more predictive of performance on CoQA and TriviaQA question answering datasets than existing baselines and can also be applied to closed-source models.

## 1 Introduction

In recent years, generative models have revolutionized daily lives well beyond the field of machine learning (Kasneci et al., 2023; Meskó & Topol, 2023). These models have found applications in diverse domains, including image creation (Ramesh et al., 2021), natural language generation (OpenAI, 2023), drug discovery (Paul et al., 2021), and speech synthesis (Ning et al., 2019). While generative models have demonstrated remarkable capabilities in generating data that closely resemble real-world samples, they often fall short in providing the vital and often overlooked aspect of uncertainty estimation (Wu & Shang, 2020). Uncertainty estimation in machine learning is a critical component of model performance assessment and deployment (Hekler et al., 2023). It addresses the inherent limitations and challenges associated with machine learning based decisions. For generative models, this may include their propensity to generate improbable or nonsensical samples ("hallucinations"). Even though uncertainty estimation methods for natural language question

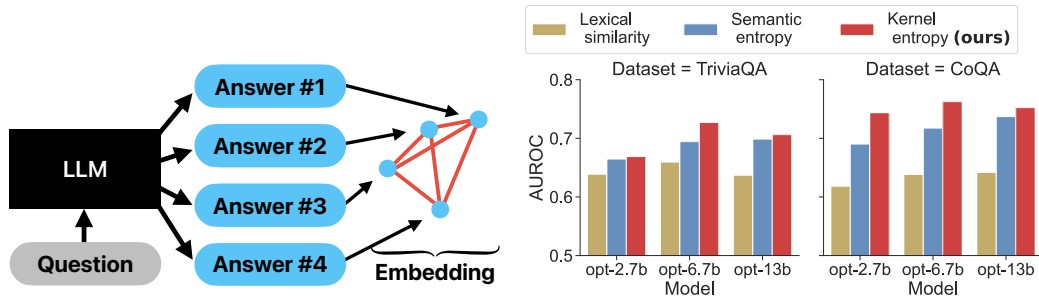

Figure 1: **Left:** Illustration of predictive kernel entropy for a generative model. A kernel measures the pairwise similarities (red lines) of outputs in a vector space. The predictive kernel entropy is then the negative average kernel value. **Right:** The predictive kernel entropy shows the best performance among uncertainty approaches for closed-source settings (c.f. Section 5.3).

answering tasks exist (Kuhn et al., 2023), they are ad-hoc without theoretical grounding and are not transferable to other data generation tasks .

Predictive uncertainty is an informal concept, but it is implied that it relates to the prediction error without requiring access to target outcomes. A formal approach to this is the bias-variance decomposition, a central concept in statistical learning theory (Bishop & Nasrabadi, 2006; Hastie et al., 2009; Murphy, 2022). It helps to understand the generalization behavior of models and naturally raises uncertainty terms by isolating the target prediction into a bias term (Gruber & Buettner, 2023). Ueda & Nakano (1996) discovered the bias-variance-covariance decomposition of the mean squared error, which is the foundation of negative correlation learning (Liu & Yao, 1999b;a; Brown, 2004) and for reducing correlations in weight averaging (Rame et al., 2022). Though the bias-variance decomposition has been generalized to distributions (Gruber & Buettner, 2023), the current theory does not include a covariance term and relies on having access to the predicted distribution. But, many generative models only indirectly fit the training distribution by learning how to generate samples. Others, such as large language models (LLMs), do explicitly fit the target distribution, but the prevalence of closed source models means the predictive distribution is often not available to the practitioner (OpenAI, 2023). This makes it infeasible to apply the powerful framework of the bias-variance decomposition in these cases.

Contrary, kernels allow to quantify differences between distributions only based on their samples without requiring access to these distributions (Gretton et al., 2012a). They are used in kernel scores to assess the goodness-of-fit for predicted distributions (Gneiting & Raftery, 2007).

As **contribution** in this work, we...

- introduce the first extension of the bias-variance-covariance decomposition beyond the mean squared error to kernel scores in Section 3, and propose unbiased and consistent estimators only requiring generated samples in Section 4.

- examine the generalisation behavior of generative models for image and audio generation and investigate how bias, variance and kernel entropy relate to the generalisation error in Section 5. This includes evidence that mode collapse of underrepresented minority groups is expressed purely in the bias.

- demonstrate how kernel entropy in combination with text embeddings outperforms existing methods for estimating the uncertainty of LLMs on common question answering datasets (c.f. Figure 1 and Section 5.3).

## 2 BACKGROUND

In this section, we give a brief introduction into kernel scores, followed up by other bias-variance decompositions and approaches for assessing the uncertainty in natural language generation.

### 2.1 KERNEL SCORES

Kernel scores are a class of loss functions for distribution predictions (Eaton, 1981; Eaton et al., 1996; Dawid, 2007). For simplicity, we omit complex-valued kernels. We refer to a symmetric kernel $k \colon \mathscr{X} \times \mathscr{X} \to \mathbb{R}$ defined on a set $\mathscr{X}$ as **positive definite** (p.d.) if $\sum_{i=1}^{n} \sum_{j=1}^{n} a_i k(x_i, x_j) a_j > 0$ for all $x_1, \ldots, x_n \in \mathscr{X}$ and $a_1, \ldots, a_n \neq 0$ with $n \in \mathbb{N}$. **Positive semi-definite** (p.s.d.) refers to the case when only '$\geq$' holds. Assume $\mathscr{P}$ is a set of distributions defined on $\mathscr{X}$ such that for a kernel $k$ the operator $\langle P \,|\, k \,|\, Q \rangle \coloneqq \int_{\mathscr{X}} \int_{\mathscr{X}} k\,(x,y)\,\mathrm{d}P\,(x)\,\mathrm{d}Q\,(y)$ is finite for all $P, Q \in \mathscr{P}$ (Eaton, 1981). It follows that $\langle . \,|\, k \,|\, . \rangle$ is a symmetric bilinear form and induces the semi-norm $\|P\|_k = \sqrt{\langle P \,|\, k \,|\, P \rangle}$. A **kernel score** $S_k \colon \mathscr{P} \times \mathscr{X} \to \mathbb{R}$ based on a p.s.d. kernel $k \colon \mathscr{X} \times \mathscr{X} \to \mathbb{R}$ is defined as (Steinwart & Ziegel, 2021)

$$S_k\,(P, y) = \|P\|_k^2 - 2\,\langle P \,|\, k \,|\, \delta_y \rangle, \tag{1}$$

where $\delta_y$ is the dirac measure at point $y$. Note that Eaton (1981) and Dawid (2007) use a slightly less general definition. If $\mathscr{P}$ only consists of Borel probability measures, then the expected kernel score $\mathbb{E}\,[S_k\,(P, Y)]$ based on a target $Y \sim Q \in \mathscr{P}$ is minimized when $P = Q$ (Gneiting & Raftery, 2007). Following Dawid (2007), we refer to $-\,\|Q\|_k^2 = \mathbb{E}\,[S_k\,(Q, Y)]$ as the **kernel entropy** function of $Q$. If $k$ is associated with a reproducing kernel Hilbert space (RKHS), then the kernel score is connected to the **maximum mean discrepancy** (MMD) via $\mathrm{MMD}_k^2\,(P, Q) = \mathbb{E}\,[S_k\,(P, Y)] + \|Q\|_k^2$

(Steinwart & Ziegel, 2021). MMDs are used for non-parametric two-sample testing (Gretton et al., 2012a) and generative image modelling (Li et al., 2015; Bińkowski et al., 2018). Compared to MMDs, kernel scores are applicable to a wider range of scenarios, since one sample of the target distribution is sufficient for evaluation. For example, MMDs cannot be computed for question-answering pairs when there is only one answer for each question in the dataset.

## 2.2 BIAS-VARIANCE (-COVARIANCE) DECOMPOSITIONS

Ueda & Nakano (1996) introduced the bias-variance-covariance decomposition for the mean squared error. For a real-valued ensemble prediction $\hat{P}^{(n)} = \frac{1}{n} \sum_{i=1}^{n} \hat{P}_i$ with identically distributed $\hat{P}_1, \ldots, \hat{P}_n$ and real-valued target $Y$ it is given by

$$\underbrace{\mathbb{E}\left[\left(\hat{P}^{(n)} - Y\right)^2\right]}_{\text{Expected Squared Error}} = \underbrace{\mathbb{V}(Y)}_{\text{Noise}} + \underbrace{(\mathbb{E}\left[\hat{P}\right] - \mathbb{E}[Y])^2}_{\text{Bias}} + \underbrace{\frac{1}{n}\mathbb{V}\left(\hat{P}\right)}_{\text{Variance}} + \underbrace{\frac{n-1}{n}\text{Cov}\left(\hat{P}, \hat{P}'\right)}_{\text{Covariance}}, \quad (2)$$

with $\hat{P} := \hat{P}_1$ and $\hat{P}' := \hat{P}_2$. Rame et al. (2022) propose an approximate bias-variance-covariance decomposition for hard-label classification but it only holds in an infinitesimal locality around the prediction. To our best knowledge, the mean squared error is the only case so far with a non-approximated decomposition. Gruber & Buettner (2023) introduced a bias-variance decomposition for loss functions of general distributions. They demonstrated that the variance term is a meaningful measure of the model uncertainty similar to confidence scores in classification. But, their formulation requires a loss-specific transformation of the distributions into a dual vector space and a covariance term is not given.

## 2.3 UNCERTAINTY IN NATURAL LANGUAGE GENERATION

In the following, we give a brief overview of uncertainty estimations in natural language generation. A common approach is predictive entropy, which is the Shannon entropy $-\int \log \hat{p}(y \mid x) \, \mathrm{d}\hat{p}(y \mid x)$ of the predicted distribution $\hat{p}$ given an input $x$ (Malinin & Gales, 2020). For a generated token sequence $\mathbf{s} = (s_1, \ldots, s_l) \in \mathbb{N}^l$ of length $l \in \mathbb{N}$ it is computed via $\sum_{i=1}^{l} \log \hat{p}(s_i \mid s_1, \ldots, s_{i-1})$, where $\hat{p}$ is the predicted distribution of the generating language model. Note that the predicted distribution is not always available for closed-source models. The computation also scales linearly with the length of the generated text, making it costly for larger text generations. Malinin & Gales (2020) propose to use length-normalisation of the predictive entropy since the Shannon entropy is systematically affected by the sequence length. Kuhn et al. (2023) propose *semantic entropy* to ease the computation of the predictive entropy by finding clusters of semantically similar generations. Another approach is *lexical similarity* (Fomicheva et al., 2020), which quantifies the average pairwise similarity between generated answers according to a similarity measure, like $\text{RougeL}$ (Lin & Och, 2004; Kuhn et al., 2023). Kadavath et al. (2022) propose the baseline *p(True)*, which asks the model itself if the generated answer is correct. Alternative approaches exist, which require an ensemble of models (Lakshminarayanan et al., 2017; Malinin & Gales, 2020). However, ensembles are practically less relevant due to the high computational cost of training even a single model.

## 3 A BIAS-VARIANCE-COVARIANCE DECOMPOSITION OF KERNEL SCORES

In this section, we state our main theoretical contribution. All proofs are presented in Appendix D. To highlight the similarity to the mean squared error case, we introduce the novel definitions for distributional variance and distributional covariance. The latter also implies a distributional correlation, which we define later in Section 4. Note that conventional variance and covariance are based on multiplication of two components ($x \cdot x$ for variance and $x \cdot y$ for covariance). We interpret $\langle . \mid k \mid . \rangle$ as a generalization of this multiplication, which directly implies the following.

**Definition 3.1.** Assume we have a p.s.d. kernel $k$ and random variables $P$ and $Q$ with outcomes in a distribution space as defined above. We define the **distributional variance** generated by $k$ of $P$ as

$$\text{Var}_k[P] = \mathbb{E}\left[\|P - \mathbb{E}[P]\|_k^2\right] \quad (3)$$

and the **distributional covariance** generated by $k$ between $P$ and $Q$ as

$$\text{Cov}_k(P, Q) = \mathbb{E}[\langle P - \mathbb{E}[P] \mid k \mid Q - \mathbb{E}[Q]\rangle]. \quad (4)$$

If $P$ is deterministic, i.e. is a random variable with only one outcome, then $\mathrm{Var}_k(P) = 0$. Further, we have $\mathrm{Cov}_k(P, P) = \mathrm{Var}_k[P]$, and, if $P$ and $Q$ are independent, then $\mathrm{Cov}_k(P, Q) = 0$. Note that the terms *kernel variance* and *kernel covariance* already exist in the literature and should not be confused with our definitions (Gretton et al., 2003).

We now have the necessary tools to state our main theoretical contribution in a concise manner.

**Theorem 3.2.** *Let $S_k$ be a kernel score based on a p.s.d. kernel $k$ and $\hat{P}$ a predicted distribution for a target $Y \sim Q$, then*

$$\underbrace{\mathbb{E}\left[S_k\left(\hat{P}, Y\right)\right]}_{\textit{Generalization Error}} = \underbrace{-\|Q\|_k^2}_{\textit{Noise}} + \underbrace{\left\|\mathbb{E}\left[\hat{P}\right] - Q\right\|_k^2}_{\textit{Bias}} + \underbrace{\mathrm{Var}_k\left(\hat{P}\right)}_{\textit{Variance}}. \tag{5}$$

*If we have an ensemble prediction $\hat{P}^{(n)} \coloneqq \frac{1}{n}\sum_{i=1}^n \hat{P}_i$ with identically distributed members $\hat{P}_1, \ldots, \hat{P}_n$, then*

$$\mathrm{Var}_k\left(\hat{P}^{(n)}\right) = \frac{1}{n}\,\mathrm{Var}_k\left(\hat{P}_1\right) + \frac{n-1}{n}\,\mathrm{Cov}_k\left(\hat{P}_1, \hat{P}_2\right). \tag{6}$$

This theorem proves the relation between a kernel-based generalization error and the distributional variance and distributional covariance. It has a wide range of practical relevance since kernels can be used for almost all data scenarios via vector embeddings (Liu et al., 2020). Consequently, we can extend the evaluation and analysis for regression, which has been done since the emergence of the mean squared error decomposition (Brown, 2004), to tasks with arbitrarily complex target distributions. This opens up possibilities for gaining new insights into the most successful generative models of recent years. In Appendix B we express the decomposition in terms of the reproducing kernel Hilbert space associated with the kernel.

In the following of this work, we use Theorem 3.2 to study the generalization behavior of generative models and to find ways to estimate the uncertainty of generated data. The presented evaluations and approaches are applicable to almost any data generation task due to the flexibility of kernels and data embeddings.

**Predictive Kernel Entropy for Single Models.** Historically, the bias-variance decomposition had a large impact on the development of some of the most established machine learning algorithms, like Random Forests (Breiman, 2001) or Gradient Boosting (Friedman, 2002). However, ensemble approaches are not similarly dominant for generative modeling. Estimating $\mathrm{Var}_k(\hat{P})$ requires an ensemble of models, which is not always feasible. Instead, note the decomposition $\mathrm{Var}_k(\hat{P}) = \mathbb{E}[\|\hat{P}\|_k^2] - \|\mathbb{E}[\hat{P}]\|_k^2$ and observe that the distributional variance depends on the predictive kernel entropy $-\|\hat{P}\|_k^2$, which is estimated for single models. The predictive kernel entropy also appears in the definition of kernel scores in Equation 1. This suggests that it may have a substantial influence on the generalization error. In Section 5, we will discover that this influence is extremely high (Pearson correlation of approx. 0.95), but the sign of the correlation is task-specific. Further, by using text embeddings, predictive kernel entropy is better than other baselines in predicting the performance of LLMs (c.f. Section 5).

## 4 UNBIASED AND CONSISTENT ESTIMATORS

If the prediction $\hat{P}$ is available in closed-form, the quantities in Theorem 3.2 can be computed according to conventional approaches (Gruber & Buettner, 2023). But, this is not the case for a lot of recently developed generative models in Deep Learning. For example, Diffusion Models (Ho et al., 2020) or closed-source LLMs (OpenAI, 2023) are also learning the training distribution, but they are often limited to generating samples. In this section, we introduce estimators of the distributional variance and covariance for the case when only samples of the distributions are available. This increases the practical applicability of Theorem 3.2 by a wide margin and allows investigating the most recent and largest generative models without constraints. We assume a minimum of two samples from each distribution is given. All estimators in the following require a two-stage sampling procedure (Särndal et al., 2003): First, distributions are sampled in an outer loop, which can be seen as clusters. In Section 5, this will be an ensemble of generative models. Second, we sample of each

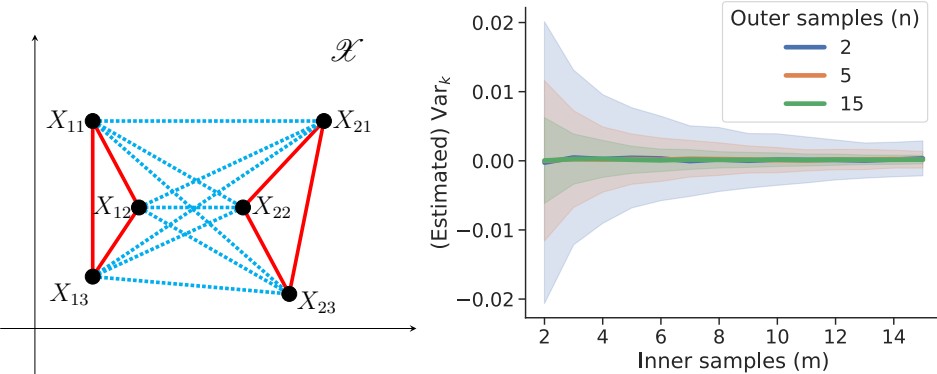

Figure 2: **Left:** Illustration of the estimator $\widehat{\mathrm{Var}}_k^{(n,m)}$ in the sample space $\mathscr{X}$ for $n = 2$ outer samples and $m = 3$ inner samples. The estimator computes the average similarity within clusters (solid red lines) minus the average similarity between clusters (dotted blue lines). Shorter lines indicate higher similarity and larger kernel values. **Right:** Estimator standard deviation for various sample sizes. Even though the estimator does not converge in theory with the inner sample size $m$, it may still be influenced significantly by it for small sample sizes.

distribution multiple times in an inner loop, which can be seen as within-cluster samples. This will be the data generations of each model.

The procedure differs slightly between the variance and covariance case. For simplicity, we also assume that all within-cluster sample sizes are the same. All estimators can be adjusted if that is not the case and will still be unbiased and consistent. Again, all proofs are presented in Appendix D.

## 4.1 DISTRIBUTIONAL VARIANCE

Assume we have a random variable $P$ with outcomes in $\mathscr{P}$ based on an unknown distribution $\mathbb{P}_P$ from which we can sample. First, we sample distributions $P_1, \ldots, P_n \overset{\text{iid}}{\sim} \mathbb{P}_P$. Then, we sample $X_{i1}, \ldots, X_{im} \overset{\text{iid}}{\sim} P_i$ for $i = 1 \ldots n$. The estimator we are about to propose is directly derived from the conventional variance estimator $\hat{\sigma}^2 := \frac{1}{n-1} \sum_{i=1}^n \left\| P_i - \frac{1}{n} \sum_{s=1}^n P_s \right\|_k^2$. Note that it holds $\hat{\sigma}^2 = \frac{1}{n} \sum_{i=1}^n \|P_i\|_k^2 - \frac{1}{n(n-1)} \sum_{i=1}^n \sum_{\substack{s=1 \\ s \neq i}}^n \langle P_i \,|\, k \,|\, P_s \rangle$, i.e. the estimator is the average of same-index pairs minus the average of the rest. Our extended estimator then uses the plug-ins $\|P_i\|_k^2 \approx \frac{1}{m(m-1)} \sum_{j=1}^m \sum_{\substack{t=1 \\ t \neq j}}^m k(X_{ij}, X_{it})$ and $\langle P_i \,|\, k \,|\, P_s \rangle \approx \frac{1}{m^2} \sum_{j=1}^m \sum_{t=1}^m k(X_{ij}, X_{st})$. The complete estimator of the distributional variance $\mathrm{Var}_k(P)$ is defined by

$$\widehat{\mathrm{Var}}_k^{(n,m)} = \underbrace{\frac{1}{nm(m-1)} \sum_{i=1}^n \sum_{j=1}^m \sum_{\substack{t=1 \\ t \neq j}}^m k(X_{ij}, X_{it})}_{\text{Average similarity within clusters}} - \underbrace{\frac{1}{n(n-1)m^2} \sum_{i=1}^n \sum_{\substack{s=1 \\ s \neq i}}^n \sum_{j=1}^m \sum_{t=1}^m k(X_{ij}, X_{st})}_{\text{Average similarity between clusters}}. \tag{7}$$

An illustration is given on the left in Figure 2. The estimator is unbiased since $\mathbb{E}[\widehat{\mathrm{Var}}_k^{(n,m)}] = \mathrm{Var}_k(P)$. Its runtime complexity is in $\mathcal{O}(m^2 n^2)$. Estimators with lower complexity, like $\mathcal{O}(mn)$, exist but are not recommendable since they have a worse performance and in most applications, generating the samples is far more costly than evaluating the estimator.

The variance of the estimator is in $\mathcal{O}\left(\frac{1}{n}\left(1 + \frac{1}{m}\right)\right)$, which proves $\widehat{\mathrm{Var}}_k^{(n,m)} \longrightarrow \mathrm{Var}_k(P)$ in probability with growing $n$ but not $m$. In words, the estimator is consistent with increasing outer samples but not inner samples. This may suggest to neglect creating inner samples and keep $m$ small, but our analysis in Appendix D.2 shows that there exist sub-terms which converge equally fast in $m$ as in $n$. In combination with the finite sample simulation in Figure 2, we recommend to use $n \approx m \geq 10$, if no prior information is available.

## 4.2 DISTRIBUTIONAL COVARIANCE AND CORRELATION

For the covariance case, assume we have random variables $P$ and $Q$ with outcomes in $\mathscr{P}$ based on an unknown joint distribution $\mathbb{P}_{PQ}$ from which we can sample. We require samples $X_{i1}, \ldots, X_{im} \overset{\text{iid}}{\sim} P_i$ and $Y_{i1}, \ldots, Y_{im} \overset{\text{iid}}{\sim} Q_i$ with $(P_1, Q_1), \ldots, (P_n, Q_n) \overset{\text{iid}}{\sim} \mathbb{P}_{PQ}$. Then, we propose the unbiased and consistent covariance estimator

$$\widehat{\text{Cov}_k}^{(n,m)}(\mathbf{X}, \mathbf{Y}) = \frac{1}{nm^2} \sum_{i=1}^{n} \sum_{j=1}^{m} \sum_{t=1}^{m} \left( k\left(X_{ij}, Y_{it}\right) - \frac{1}{n-1} \sum_{\substack{s=1 \\ s \neq i}}^{n} k\left(X_{ij}, Y_{st}\right) \right) \tag{8}$$

with $\mathbf{X} \coloneqq (X_{ij})_{i=1\ldots n, j=1\ldots m}$ and $\mathbf{Y} \coloneqq (Y_{ij})_{i=1\ldots n, j=1\ldots m}$. It has the same runtime complexity and convergence rate as the variance estimator of Equation 7 (c.f. Appendix D.3). While the distributional covariance is directly implied by Theorem 3.2, it is difficult to interpret since it is not bounded. Consequently, we propose the distributional correlation estimator based on Equation 8 given by

$$\widehat{\text{Corr}_k}^{(n,m)} = \frac{\widehat{\text{Cov}_k}^{(n,m)}(\mathbf{X}, \mathbf{Y})}{\sqrt{\widehat{\text{Cov}_k}^{(n,m)}(\mathbf{X}, \mathbf{X})}\sqrt{\widehat{\text{Cov}_k}^{(n,m)}(\mathbf{Y}, \mathbf{Y})}} \in [-1, 1]. \tag{9}$$

It is consistent since continuous transformations of consistent estimators are also consistent (Shao, 2003), i.e. for $n \to \infty$ in probability

$$\widehat{\text{Corr}_k}^{(n,m)} \longrightarrow \text{Corr}_k(P, Q) \coloneqq \frac{\text{Cov}_k(P, Q)}{\sqrt{\text{Var}_k(P)}\sqrt{\text{Var}_k(Q)}}. \tag{10}$$

We use the covariance estimator for the variance terms since the variance estimator can be negative and the correlation estimator is only asymptotically unbiased no matter the choice.

In the next section, we show that distributional correlation, as implied by Theorem 3.2, is a natural tool to gain new insights into the fitting process of generative models. For example, the correlations between epochs indicate how stable the convergence during training is.

## 5 APPLICATIONS

In this section, we apply the proposed statistical tools to assess generative models across a variety of different data generation tasks. Specifically, we put emphasis on instance-level uncertainty estimation. A meaningful measure of uncertainty is able to predict the loss for a given prediction. We use the Pearson correlation coefficient to quantify how well an uncertainty measure predicts a continuous loss, and the area under receiver operator characteristic (AUROC) for a binary loss. We first start in Section 5.1 with diffusion models for image generation on the synthetic InfiMNIST dataset for a detailed examination of their generalization behavior. In Section 5.2, we evaluate an ensemble of Glow-TTS models for text-to-speech synthesis on the SpeechLJ dataset. In all cases, kernel entropy shows strong performance as uncertainty measure. Last, we use kernel entropy to outperform other baselines in uncertainty estimation for natural language generation in Section 5.3. There, we evaluate single OPT models of different sizes on the question answering datasets CoQA and TriviaQA. The source code of all experiments is openly available at `https://github.com/[waiting-for-acceptance]`.

## 5.1 IMAGE GENERATION

For image generation, we use conditional diffusion models (Ho et al., 2020; Ho & Salimans, 2021) trained on MNIST-like datasets. We use InfiMNIST to sample an infinite number of uniquely perturbed MNIST images (Loosli et al., 2007). By simulating the data generation process we can assess the ground truth generalization error of the diffusion model. We sample $n = 20$ distinct training sets from InfiMNIST each of size 60.000 (similar as MNIST). We then train a model on each training set. This is in correspondance to how generalization error, bias, and variance are evaluated for regression and classification tasks (Ueda & Nakano, 1996; Gruber & Buettner, 2023). We use $m = 20$ generated images per class and per model for all estimators. The predictive kernel entropy is only evaluated on a single model to stay as closely as possible to practical constraints. Our kernel choice is the commonly

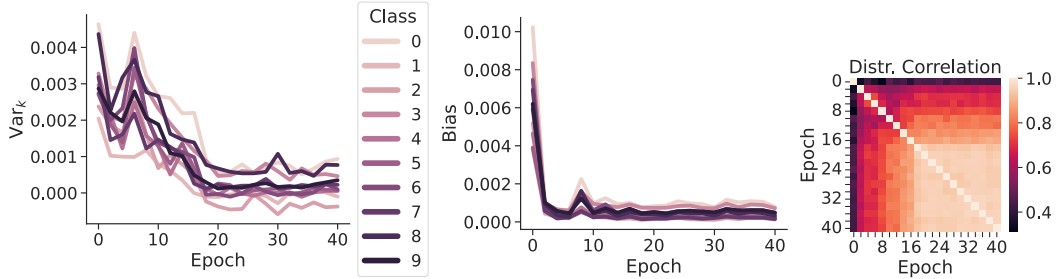

Figure 3: **Left:** The variance starts high and is reduced throughout training. From 20 epochs onwards, the variance stays stable for all classes and no overfitting can be observed. **Mid:** The bias is reduced a lot quicker than the variance, reaching its minimum at 5 epochs, and converges after 10 epochs. **Right:** The distributional correlation between training epochs shows similar to the variance that convergence happens around epoch 20. Remarkably, the 'square' of very high correlations indicates that the model is stable in its convergence and does not iterate through equally good solutions.

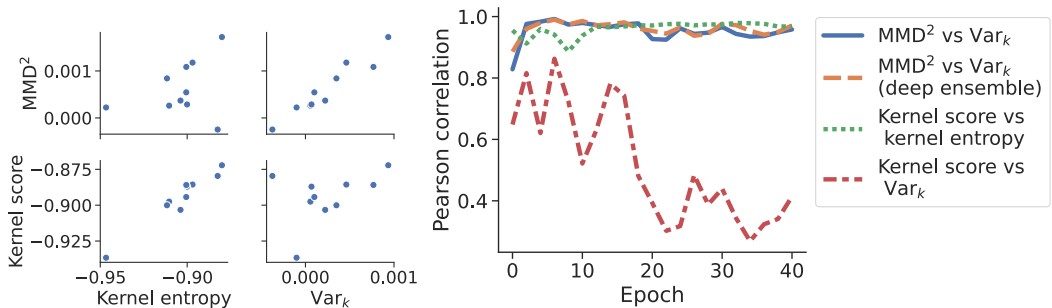

Figure 4: **Left:** Dependence between squared MMD and distributional variance. The distributional variance correlates linearly with the squared MMD. **Right:** Pearson correlation between squared MMD and distributional variance is very high ($\approx 0.95$) throughout training. Approximation via deep ensembles does not deteriorate this relation. Consequently, distributional variance and kernel entropy represent viable measure of uncertainty.

used RBF kernel $k_{\mathrm{rbf}}(x, y) = \exp(-\gamma \|x - y\|_2^2)$, where $x$ and $y$ are flattened images and $\gamma$ a normalization factor based on the number of pixels (Schölkopf, 1997; Schölkopf & Smola, 2002; Han et al., 2012; Gretton et al., 2012a; Li et al., 2015; Bińkowski et al., 2018; Liu et al., 2020). We first analyse the generalization behavior and then assess different approaches for uncertainty estimation.

In Figure 3, we plot the distributional variance, bias, and distributional correlation throughout training for every second epoch. As can be seen, the model converges quicker for the bias than the variance (around epochs 10 and 20). Further, no overfitting occurs since both variance and bias stay small. The correlation matrix also shows convergence around epoch 20, but, more interestingly, it shows a square of high correlations in the lower right corner. This is an indication that the diffusion model training is stable in its convergence and does not iterate through different minima in the optimization landscape.

In Figure 4, we compare the relations between kernel score and $\mathrm{MMD}^2$ to distributional variance and predictive kernel entropy for each class. As can be seen, the predictive kernel entropy correlates strongly linearly with the generalization error (kernel score) but not so much with the generalization discrepancy ($\mathrm{MMD}^2$), while the distributional variance correlates strongly linearly with the $\mathrm{MMD}^2$ but not the kernel score. This corre-

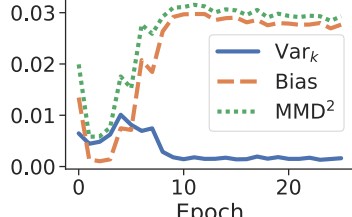

Figure 5: $\mathrm{MMD}^2$, variance, and bias for class '0' throughout training with reduced training set of '0's. After 5 epochs, mode collapse occurs, which is only expressed in the increased bias. This indicates, that mode collapse is a contrary phenomenon to overfitting.

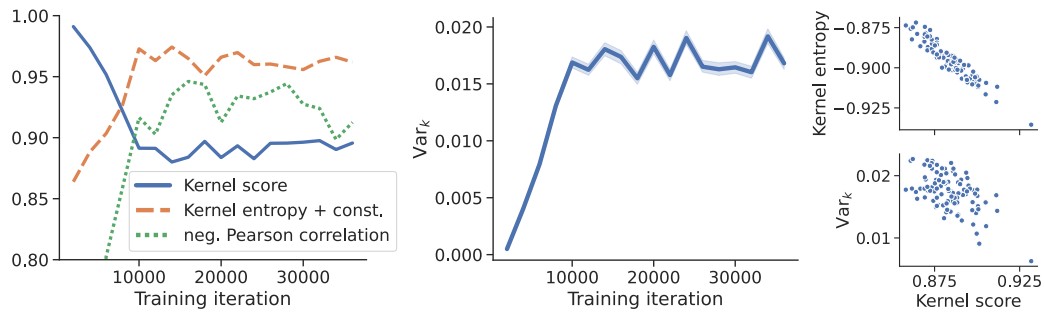

Figure 6: **Left:** Kernel score, predictive kernel entropy, and their Pearson correlation throughout training for Glow-TTS. The entropy indicates that the model initially predicts too narrow distributions, which widen until convergence at around 10000 iterations. After convergence, the correlation between predictive entropy and kernel score is very high. **Mid:** The variance of an Deep Ensemble is also initially very small and converges at the same time as kernel score and entropy. **Right:** Comparing kernel score and kernel entropy as well as variance for 100 test instances at 16000 training iterations. Again, the correlation shows strong linearity.

lation can be observed throughout the whole training and gives
a very high Pearson correlation coefficient of around $0.95$. Importantly for practical settings, the correlation between $MMD^2$ and distributional variance does not deteriorate when we use a deep ensemble trained on a single dataset.
In summary, these results demonstrate that both distributional variance as well as predictive kernel entropy are viable measure of uncertainty to predict the correctness of generated instances, either in terms of kernel score or $MMD^2$.

We next set out to use our estimators for bias and variance to elucidate the phenomenon of mode collapse. When generative models are used to learn the distribution of a given training set, there are often groups of different sizes present. In these cases, a common occurence is mode collapse towards the majority groups, i.e. the model catastrophically fails to model the minority groups. To simulate this scenario in our setting, we repeat our evaluation but reduce the frequency of images of digit '0' to $\approx 1\%$ in each training set. The bias-variance curves of digit '0' across training (Figure 5) reveals the expected mode collapse, and, most importantly, demonstrates that it is only expressed in terms of the bias. The variance term is further reduced throughout training as if no collapse occured. This suggests that mode collapse may be seen as a contrary phenomenon to overfitting. While in overfitting, the variance increases and the bias reduces, this is vice versa for mode collapse for prolonged training.

## 5.2 AUDIO GENERATION

We next evaluate the fitting and generalization behavior of the generative flow model Glow-TTS (Kim et al., 2020) on the text-to-speech dataset LJSpeech (Ito & Johnson, 2017) throughout training (Appendix C). We train a deep ensemble via $n = 10$ different weight initializations on $90\%$ of the available data. We evaluate the models every 2000 training iterations by generating $m = 10$ speech waveforms for each of 100 test instances. The evaluation includes the kernel score, kernel entropy and the distributional variance. Note that ensemble members are only required to compute the distributional variance – we compute kernel entropy for a single model, as before. Here, we use the Laplacian kernel $k_{\text{lap}}(x, y) = \exp(-\gamma \|x - y\|_1)$ (Schölkopf & Smola, 2002), where $x$ and $y$ are waveforms represented by vectors and $\gamma$ a normalization constant based on the waveform length. The results are depicted in Figure 6. Our analyses reveal that similar as for image generation, overfitting does not occur for prolonged training and the Pearson correlation between kernel entropy and kernel score is very high after convergence. But, the entropy is initially very small and increases until convergence. This indicates that a successful training requires to widen the predicted distribution. Consequently, the correlation between kernel entropy and kernel score is negative, since badly fitted instances have a more narrow predicted distribution. We also conducted the evaluations with the RBF kernel, which gives similar but slightly more erratic curves than the Laplacian kernel (Appendix C).

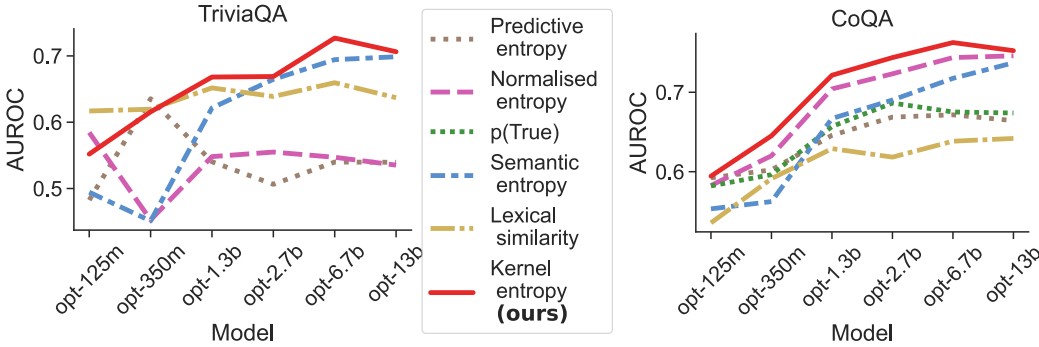

Figure 7: Area-Under-Curve of answer accuracy based on thresholds for different uncertainty measures. The kernel entropy outperforms other baselines in predicting the correctness of generated answers across a wide range of differently sized models.

## 5.3 NATURAL LANGUAGE GENERATION

An instance-level uncertainty measure is supposed to predict the correctness of an individual prediction and should therefore be highly correlated to the loss. In all experiments so far, we observed a very high correlation between kernel entropy and kernel score. This indicates that kernel entropy is an excellent measure of uncertainty. In the following, we examine kernel entropy to predict the correctness of LLMs on question answering datasets. Here, the setup differs from the previous experiments by two aspects.

First, we follow Kuhn et al. (2023) and do not use a kernel score but a binarized version of the RougeL similarity as loss. For two sequences $s, t$, it is defined as $\text{RougeL}(s, t) = \frac{2}{\text{length}(s)+\text{length}(t)} \text{LCS}(s, t)$, where LCS is the length of the longest common sequence between its two inputs (Lin & Och, 2004). Kuhn et al. (2023) propose to use the binary loss $L(answer, target) = \mathbb{1}\{\text{RougeL}(answer, target) > 0.3\}$. This turns predicting the loss value into a binary classification problem. Consequently, the AUROC is more meaningful than Pearson correlation for evaluating the performance of uncertainty measures (Kadavath et al., 2022).

Second, we do not directly use the generated answers as inputs for a kernel but, instead, their vector embeddings. A well-trained vector embedder maps text into a semantically meaningful vector space in which a kernel then measures similarities (Camacho-Collados & Pilehvar, 2018).

The investigated uncertainty baselines are predictive entropy, normalised (predictive) entropy, p(True), semantic entropy, and lexical similarity (Kuhn et al., 2023). We consider uncertainty estimation for question answering predictions of the datasets CoQA (Reddy et al., 2019) with 7983 test instances and TriviaQA (Joshi et al., 2017) with 5383 test instances. We use OPT models (Zhang et al., 2022) of all available sizes except the 30 billion parameter version, which is computationally prohibitive. For our kernel entropy, we use the RBF kernel and text embeddings computed via a pretrained e5-small-v2 (Wang et al., 2022). The results are depicted in Figure 7. As can be seen, kernel entropy is the most robust approach and outperforms other baselines for uncertainty estimation in almost all cases. We can achieve further improvements in our approach when we use alternative embedders (c.f. Appendix C). The cosine similarity, which is used in natural language processing (Steinbach et al., 2000), and other kernels show similar results as the RBF kernel in Appendix C.

## 6 CONCLUSION

In this work we introduced the first bias-variance-covariance decomposition beyond the mean squared error for kernel scores. We proposed estimators for the variance and covariance terms which only require samples of the predictive distributions but not the distributions themselves. This allows to evaluate all terms in the composition for arbitrary generative models and even in the closed-source setting. We studied empirically the fitting behavior of common models for image and audio generations, and demonstrated that kernel entropy and variance are viable measures of uncertainty. Finally, we showed that kernel entropy outperforms other baselines for predicting the correctness of LLMs in question answering tasks.

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

# A  OVERVIEW

In the following, we offer further details and additional experiments in Appendix C, and we provide all missing proofs in Appendix D.

# B  DECOMPOSITION IN REPRODUCING KERNEL HILBERT SPACES

The literature on MMD expanded to a significant size in the last decade (Gretton et al., 2012a;b; Chwialkowski et al., 2016; Liu et al., 2020; Kübler et al., 2020; Shekhar et al., 2022; Schrab et al., 2022; 2023). The MMD is usually used in the context of reproducing kernel Hilbert spaces (RKHS). In the following, we express Theorem 3.2 according to RKHS and MMD to offer an alternative perspective on our result. Assume the kernel $k$ is associated with an RKHS $\mathscr{H}$ with inner product $\langle .,. \rangle_{\mathscr{H}}$ and norm $\| . \|_{\mathscr{H}}$. The norm based on $k$ in the distribution space relates to the RKHS norm via $\|Q\|_k = \|\mu_Q\|_{\mathscr{H}}$ with mean embedding $\mu_Q := \mathbb{E}\left[k\left(Y,.\right)\right] \in \mathscr{H}$ for a $Y \sim Q \in \mathscr{P}$. Consequently, given a prediction $\hat{P}$ we have

$$\underbrace{\mathbb{E}\left[\mathrm{MMD}_k^2\left(\hat{P}, Q\right)\right]}_{\text{Generalization Discrepancy}} = \underbrace{\left\|\mathbb{E}\left[\mu_{\hat{P}}\right] - \mu_Q\right\|_{\mathscr{H}}^2}_{\text{Bias}} + \underbrace{\mathbb{E}\left[\left\|\mu_{\hat{P}} - \mathbb{E}\left[\mu_{\hat{P}}\right]\right\|_{\mathscr{H}}^2\right]}_{\text{Variance}}. \quad (11)$$

The covariance decomposition can be expressed similarly since $\langle P \mid k \mid Q \rangle = \langle \mu_P, \mu_Q \rangle_{\mathscr{H}}$. Note that the bias and variance terms in Theorem 3.2 and Equation 11 are equal.

# C  EXTENDED EXPERIMENTS

In this section, we give more details on the experiments and show further results.

## C.1  EXPERIMENTAL DETAILS

We give some additional details on the experimental setup. First, we formalize computing the kernel entropy in Algorithm 1. Second, we describe in more detail the image, audio, and natural language generation experiments.

---

**Algorithm 1** Uncertainty estimation via kernel entropy

---

**Require:** Generated outputs $a_1, \ldots, a_n$ for a given input, p.s.d. kernel $k$, (optional) embedder $\phi$
    **if** $\phi$ is given **then**
        **for** $i \in \{1, \ldots, n\}$ **do**
            $a_i \leftarrow \phi\left(a_i\right)$
        **end for**
    **end if**
    **return** $-\frac{1}{n(n-1)} \sum_{i=1}^n \sum_{j \neq i}^n k\left(a_i, a_j\right)$         ▷ Estimated kernel entropy

---

### C.1.1  IMAGE GENERATION

We use the following procedure for the simulation in Figure 2. First, we sample 32 distinct training sets from InfiMNIST of size 60.000. Then, we train a conditional diffusion model on each training set for 20 epochs. We adopt implementation and hyperparameters from open source PyTorch code of a conditional diffusion model trained on normal MNIST (Paszke et al., 2019). This includes an initial learning rate of 1e-4, a batch size of 256, 400 diffusion steps, and a feature dimension of 128. We then generate 100 images of class '0' of each model after training. Finally, to get the approximate standard deviation of each tick and each line in Figure 2, we estimate the distributional variance 1000 times on randomly drawn samples without replacement of all generated images. The normalization constant in the RBF kernel is set to $\gamma = \frac{1}{728}$ since each image has 728 pixels. We also repeated the whole procedure for other classes with similar results.

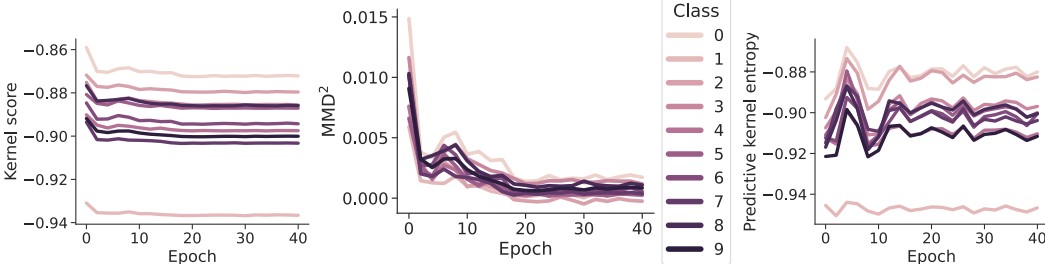

Figure 8: **Left:** Kernel score throughout training. The kernel score cannot be compared meaningfully between classes since each optimum depends a constant specific to each target class. **Mid:** $MMD^2$ throughout training. Here, it is easier to compare the errors since the $MMD^2$ is zero for the optimal prediction. **Right:** The predictive kernel entropy of each class fluctuates throughout training but stays constant upon convergence.

The results in Figure 3 and 4 are produced in a similar manner. Only difference here is that we train on only 20 training sets for 40 epochs and generate only 20 images per class. We chose these numbers based on the insights gained by the previous simulation experiment. All training was done on Nvidia RTX5000 GPUs.

### C.1.2 AUDIO GENERATION

For the audio experiments, we use an implementation of Glow-TTS given in the TTS library (Eren & The Coqui TTS Team, 2021). The LJSpeech dataset consists of 13.100 instances of text-speech pairs. Each speech is of a single woman reading out loud the corresponding text. We use a random 90% of the data for training and a batch size of 32. On this single training set, we train 12 randomly initialized models for 100 epochs with an initial learning rate of 1e-2. The evaluation happens every 2.000 gradient descent iterations for each model. For a single model in a single evaluation step, we generate 10 waveforms (speeches) for each of 100 test instances. The normalization constant in the Laplacian and RBF kernel is set to $\gamma = \frac{1}{\lambda_{\text{iter,inst}}}$, where $\lambda_{\text{iter,inst}}$ is the longest generated waveform in each evaluation step $\text{iter}$ and each test instance $\text{inst}$. The generated audio instances are of various length, so we pad them with zeros to match their length.
Further, all training was done on Nvidia RTX5000 GPUs. But, noteworthy to this experiment, storing the model iterations and generated waveforms required up to 400 GB of hard disk storage.

### C.1.3 NATURAL LANGUAGE GENERATION

For the natural language experiments, we adopted the experimental setup of Kuhn et al. (2023). We used their provided code implementations including the hyperparameters. This includes a temperature of $T = 0.5$ for generating the answers used for uncertainty estimation. Similarly, we use 10 answer generations for each prompt for TriviaQA and 20 answer generations for CoQA. All natural language models are pretrained and downloaded from HuggingFace (Wolf et al., 2020). We used a single Nvidia A6000 GPU for the natural language experiments.

### C.2 ADDITIONAL RESULTS

In the following, we give some additional results for image, audio, and language generation.

### C.2.1 IMAGE GENERATION

We start with the image experiments. In Figure 8, we show the corresponding generalization error (kernel score), $MMD^2$, and predictive kernel entropy values throughout training of the same setup as in Figure 3. As can be seen, $MMD^2$ is more interpretative for comparing different classes. But, the MMD can also not be evaluated in a lot of practical cases including the audio and natural language settings in this work. Further, the kernel entropy does not show a trend throughout training contrary to the audio setting seen in Figure 6.

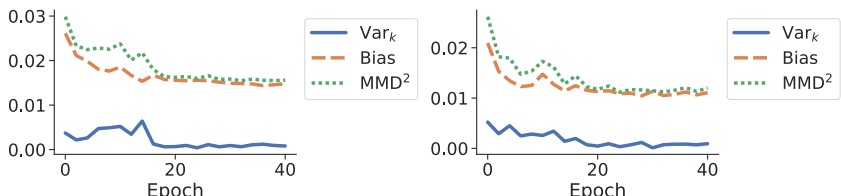

Figure 9: Generalization curves of the digit, which we reduce in frequency during training of DDPMs on InfiMNIST. The reduced digit has only approx. 60 training instances while the other digits have approx. 6000 each. **Left:** We reduce and evaluate exclusively digit '2'. **Right:** We reduce and evaluate exclusively digit '3'. We cannot observe a mode collapse as with digit '0' in Figure 5. The bias is still elevated compared to normal digit frequencies (c.f. Figure 3).

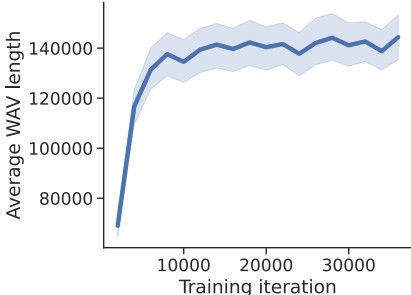

Figure 10: Average WAV length of generated audio instances by Glow-TTS. Initially, the model generates only very short instances, which explains the low initial kernel entropy.

Further, we perform similar experiments as in Figure 5, where we observed a mode collapse. In Figure 9, we repeat the evaluation but use digit '2' and digit '3' instead of digit '0'. As can be seen via our bias-variance decomposition, mode collapse does not occur, but the bias term is still larger than in Figure 3. This underlines that our decomposition is a useful tool to investigate and analyze the fitting behaviour of generative models.

### C.2.2 AUDIO GENERATION

We repeat the evaluations of the audio generations in Section 5.2, where we used the Laplacian kernel, with the RBF kernel. It is known that the RBF kernel does not scale well to higher dimensions (Bińkowski et al., 2018). We represent the generated audio instances via vectors of various lengths, often exceeding 100,000 dimensions (c.f. Figure 10).

Consequently, we expect the RBF kernel to behave more erratic than the Laplacian kernel in the main paper. The results are shown in Figure 11.

### C.2.3 NATURAL LANGUAGE GENERATION

Next, we continue with the natural language experiments.
In Figure 12, we confirm that the correlation between the predictive kernel entropy and the RougeL (which is supposed to be maximized) has the same sign as in the image experiments in Figure 4.

We also evaluate different embedders. For this, we compare different ones, which have been pretrained on a variety of different training sets and with different embedding dimensions available on HuggingFace. These are e5-small-v2 (384 dimensions; used in the main paper) (Wang et al., 2022), gte-large (1024 dimensions) (Li et al., 2023), all-mpnet-base-v2 (768 dimensions) (Song et al., 2020), as well as all-MiniLM-l6-v2 and all-MiniLM-L12-v2 (both 384 dimensions) (Wang et al., 2020). The models all-mpnet-base-v2, all-MiniLM-l6-v2, and all-MiniLM-L12-v2 included the training set of TriviaQA in their training data, while e5-small-v2 and gte-large did not. Consequently, these three models are an unfair comparison for TriviaQA to other baselines not using its training set. In Figure 13, we compare the ability of each embedder in combination with the RBF kernel to predict

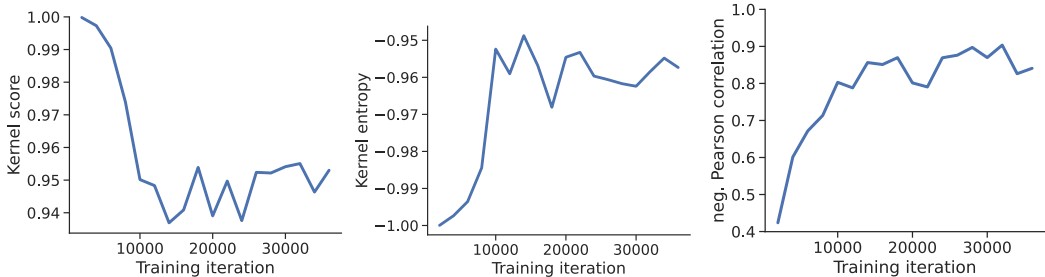

Figure 11: Audio generation results for kernel score, kernel entropy, and negative Pearson correlation between them with the RBF kernel. The trends are similar as in Figure 6 but more erratic and the absolute correlation is slightly less.

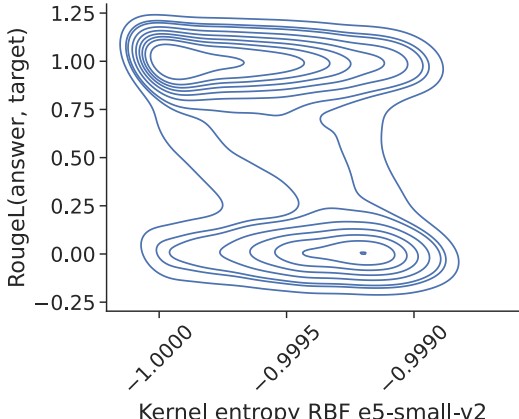

Figure 12: Kernel density estimation of kernel entropy and RougeL between answer and target for opt-1.3b model on CoQA. A large RougeL corresponds to a high accuracy and a low error. Consequently, low kernel entropy indicates a high likelihood of answer correctness.

the answer accuracy for the same settings as in Figure 7 in the main paper. As we can see for CoQA, all embedders provide approximately similar performance. This indicates that kernel entropy is a robust approach as long as the embedder is meaningful. The results for TriviaQA are similar, but all-MiniLM-L6-v2 and all-MiniLM-L12-v2 perform comparably better. We hypothesis that this is due to them being trained on the training set of TriviaQA. This suggests that we can achieve even better uncertainty estimates by using a task specific training set.

We also compare the impact of different kernel choices in Figure 14. The differences are marginal for cosine similarity, and RBF and polynomial kernel. Contrary, the Laplacian kernel performs often worse. The lack of a difference between RBF or polynomial kernel and cosine similarity is surprising considering that e5-small-v2 has been trained using the cosine similarity (Wang et al., 2022). Our results suggests that the embedder has substantially more influence than the kernel, and that resources should be spent on optimizing the former and not the latter.

Last, we compare different number of generated answers to estimate the kernel entropy in the case of Opt-13b on CoQA in Figure 15. As can be seen, more samples are continuously better. Consequently, we expect to get even better results in Figure 7 for larger numbers of generations.

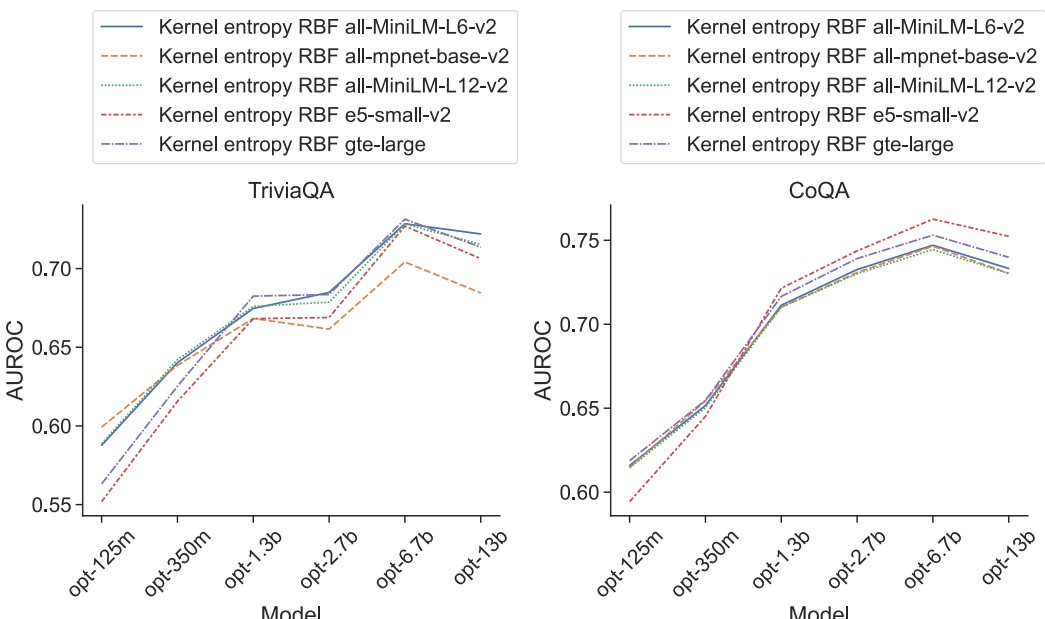

Figure 13: Comparing uncertainty estimates of kernel entropy for different embedders and RBF kernel. In both cases, the differences are not substantial. But, all-MiniLM-L6-v2 and all-MiniLM-L12-v2 perform comparably better on TriviaQA, likely due to them being trained on the TriviaQA training set.

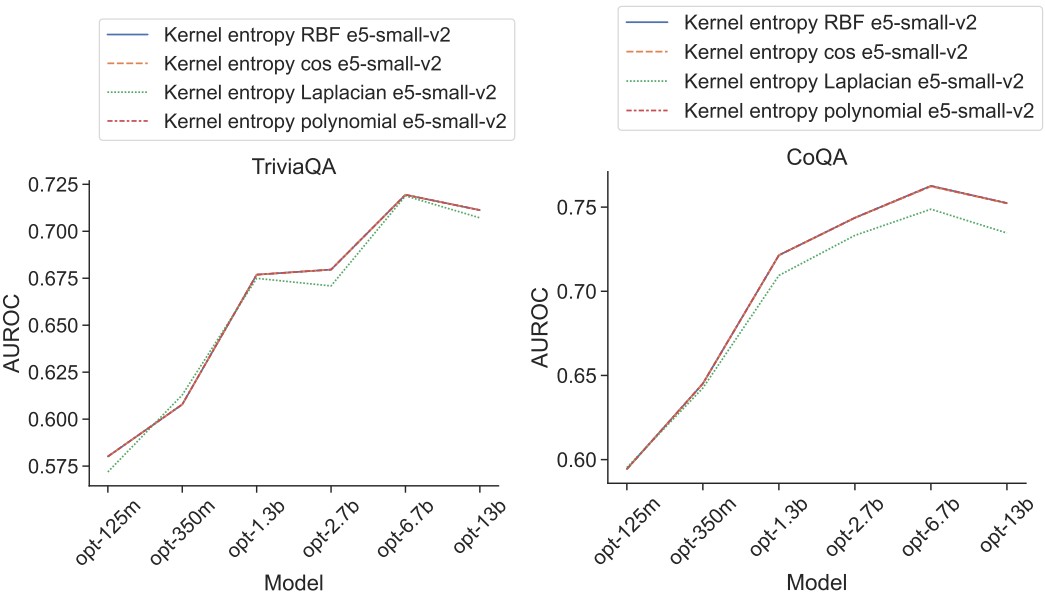

Figure 14: Comparing uncertainty estimates of kernel entropy for cosine similarity and RBF, polynomial, and Laplacian kernel. Even though the embedder e5-small-v2 is trained via cosine similarity, the performance differences are marginal.

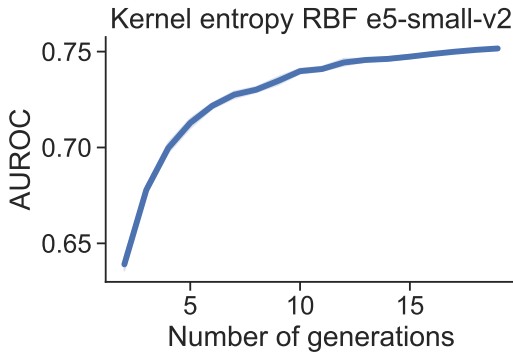

Figure 15: Number of generated answers to estimate kernel entropy compared to the AUROC for Opt-13b on CoQA. More generations monotonically improve kernel entropy as an uncertainty estimate.

## D  MISSING PROOFS

In this section, we give all missing proofs for Theorem 3.2 and for the statements in Section 4. We first start with the proof for the bias-variance-covariance decomposition in Section D.1. Then in Section D.2, we solve the expectation and the variance of the distributional variance estimator proposed in Equation 7. Last, we do the same for the distributional covariance estimator of Equation 8 in Section D.3.

### D.1  BIAS-VARIANCE-COVARIANCE DECOMPOSITION

In Theorem 3.2, the covariance decomposition is introduced after stating the more simpler bias-variance decomposition. Here, we prove both in one go. Assume we have target $Y \sim Q \in \mathscr{P}$ and ensemble prediction $\hat{P}^{(n)} := \frac{1}{n} \sum_{i=1}^{n} \hat{P}_i$ of $n \in \mathbb{N}$ identically distributed predictions $\hat{P}_1, \ldots, \hat{P}_n$ with outcomes in $\mathscr{P}$. Further, assume that all expectations in the following are finite. We will use $\hat{P} := \hat{P}_1$ and $\hat{P}' := \hat{P}_2$. Then, the decomposition can be constructed via

$$\mathbb{E}\left[-S_k\left(\hat{P}^{(n)}, Y\right)\right]$$

$$= \mathbb{E}\left[\left\|\hat{P}^{(n)}\right\|_k^2 - 2\left\langle\hat{P}^{(n)}\,\middle|\,k\,\middle|\,\delta_Y\right\rangle\right]$$

$$\overset{(i)}{=} \mathbb{E}\left[\left\|\hat{P}^{(n)}\right\|_k^2 - 2\left\langle\hat{P}^{(n)}\,\middle|\,k\,\middle|\,Q\right\rangle\right]$$

$$= \mathbb{E}\left[\left\|\hat{P}^{(n)}\right\|_k^2\right] - 2\left\langle\mathbb{E}\left[\hat{P}^{(n)}\right]\,\middle|\,k\,\middle|\,Q\right\rangle$$

$$= \mathbb{E}\left[\left\|\hat{P}^{(n)}\right\|_k^2\right] - 2\left\langle\mathbb{E}\left[\hat{P}^{(n)}\right]\,\middle|\,k\,\middle|\,Q\right\rangle + 2\left\|\mathbb{E}\left[\hat{P}^{(n)}\right]\right\|_k^2 - 2\mathbb{E}\left[\left\langle\mathbb{E}\left[\hat{P}^{(n)}\right]\,\middle|\,k\,\middle|\,\hat{P}^{(n)}\right\rangle\right]$$

$$= \mathbb{E}\left[\left\|\hat{P}^{(n)} - \mathbb{E}\left[\hat{P}^{(n)}\right]\right\|_k^2\right] - 2\left\langle\mathbb{E}\left[\hat{P}^{(n)}\right]\,\middle|\,k\,\middle|\,Q\right\rangle + \left\|\mathbb{E}\left[\hat{P}^{(n)}\right]\right\|_k^2$$

$$= \mathbb{E}\left[\left\|\hat{P}^{(n)} - \mathbb{E}\left[\hat{P}^{(n)}\right]\right\|_k^2\right] - 2\left\langle\mathbb{E}\left[\hat{P}^{(n)}\right]\,\middle|\,k\,\middle|\,Q\right\rangle + \left\|\mathbb{E}\left[\hat{P}^{(n)}\right]\right\|_k^2 + \|Q\|_k^2 - \|Q\|_k^2$$

$$= -\|Q\|_k^2 + \left\|\mathbb{E}\left[\hat{P}^{(n)}\right] - Q\right\|_k^2 + \mathbb{E}\left[\left\|\hat{P}^{(n)} - \mathbb{E}\left[\hat{P}^{(n)}\right]\right\|_k^2\right]$$

$$\overset{(ii)}{=} -\|Q\|_k^2 + \left\|\mathbb{E}\left[\hat{P}\right] - Q\right\|_k^2 + \mathbb{E}\left[\left\|\hat{P}^{(n)} - \mathbb{E}\left[\hat{P}\right]\right\|_k^2\right] \tag{12}$$

$$= -\|Q\|_k^2 + \left\|\mathbb{E}\left[\hat{P}\right] - Q\right\|_k^2 + \mathbb{E}\left[\left\|\frac{1}{n}\sum_{i=1}^n \hat{P}_i - \mathbb{E}\left[\hat{P}\right]\right\|_k^2\right]$$

$$= -\|Q\|_k^2 + \left\|\mathbb{E}\left[\hat{P}\right] - Q\right\|_k^2 + \frac{1}{n^2}\sum_{i,j=1}^n \mathbb{E}\left[\left\langle\hat{P}_i - \mathbb{E}\left[\hat{P}\right]\,\middle|\,k\,\middle|\,\hat{P}_j - \mathbb{E}\left[\hat{P}\right]\right\rangle\right]$$

$$= -\|Q\|_k^2 + \left\|\mathbb{E}\left[\hat{P}\right] - Q\right\|_k^2 + \frac{1}{n^2}\sum_{i,j=1}^n \mathrm{Cov}_k\left(\hat{P}_i, \hat{P}_j\right)$$

$$= -\|Q\|_k^2 + \frac{1}{n^2}\sum_{i=1}^n \mathrm{Var}_k\left(\hat{P}_i\right) + \frac{1}{n^2}\sum_{\substack{i,j=1\\i\neq j}}^n \mathrm{Cov}_k\left(\hat{P}_i, \hat{P}_j\right) + \left\|\mathbb{E}\left[\hat{P}\right] - Q\right\|_k^2$$

$$\overset{(ii)}{=} -\|Q\|_k^2 + \left\|\mathbb{E}\left[\hat{P}\right] - Q\right\|_k^2 + \frac{1}{n^2}n\,\mathrm{Var}_k\left(\hat{P}\right) + \frac{1}{n^2}n(n-1)\,\mathrm{Cov}_k\left(\hat{P}, \hat{P}'\right)$$

$$= \underbrace{-\|Q\|_k^2}_{\text{Noise}} + \underbrace{\left\|\mathbb{E}\left[\hat{P}\right] - Q\right\|_k^2}_{\text{Bias}} + \underbrace{\frac{1}{n}\mathrm{Var}_k\left(\hat{P}\right)}_{\text{Variance}} + \underbrace{\frac{n-1}{n}\mathrm{Cov}_k\left(\hat{P}, \hat{P}'\right)}_{\text{Covariance}}$$

(i) $Y$ and $\hat{P}^{(n)}$ independently distributed
(ii) $\hat{P}_1, \ldots, \hat{P}_n$ identically distributed

## D.2 DISTRIBUTIONAL VARIANCE ESTIMATOR

In general, the notation of Eaton (1981) allows us to write for random variables $P$ and $Q$ with outcomes in a distribution space, and for independent $X \sim P$ and $Y \sim Q$ that

$$\mathbb{E}\left[k\left(X, Y\right) \mid P, Q\right] = \int k\left(x, y\right) \mathrm{d}P \otimes Q\left(x, y\right) = \int \int k\left(x, y\right) \mathrm{d}P\left(x\right) \mathrm{d}Q\left(y\right) = \left\langle P\,\middle|\,k\,\middle|\,Q\right\rangle, \tag{13}$$

where we used Tonelli's Theorem to split the integral.

Note we have for $j \neq t$ that $X_{ij}$ and $X_{it}$ are independent given $P_i$ for all $i = 1, \ldots, n$. From this follows

$$\mathbb{E}\left[k\left(X_{ij}, X_{it}\right)\right] = \mathbb{E}\left[\mathbb{E}\left[k\left(X_{ij}, X_{it}\right) \mid P_i\right]\right]$$
$$\stackrel{\text{iid}}{=} \mathbb{E}\left[\mathbb{E}\left[k\left(X_{i1}, X_{i2}\right) \mid P_i\right]\right]$$
$$\stackrel{\text{iid}}{=} \mathbb{E}\left[\langle P_i \mid k \mid P_i \rangle\right]$$
$$\stackrel{\text{iid}}{=} \mathbb{E}\left[\langle P \mid k \mid P \rangle\right] \tag{14}$$

as well as for $i \neq s$

$$\mathbb{E}\left[k\left(X_{ij}, X_{st}\right)\right] = \mathbb{E}\left[\mathbb{E}\left[k\left(X_{ij}, X_{st}\right) \mid P_i, P_s\right]\right]$$
$$\stackrel{\text{iid}}{=} \mathbb{E}\left[\mathbb{E}\left[k\left(X_{i1}, X_{s1}\right) \mid P_i, P_s\right]\right]$$
$$\stackrel{\text{iid}}{=} \mathbb{E}\left[\langle P_i \mid k \mid P_s \rangle\right]$$
$$\stackrel{\text{iid}}{=} \langle \mathbb{E}\left[P\right] \mid k \mid \mathbb{E}\left[P\right] \rangle. \tag{15}$$

### D.2.1 EXPECTATION OF THE ESTIMATOR

Using these equations gives

$$\mathbb{E}\left[\widehat{\mathrm{Var}}_k^{(n,m)}\right]$$

$$= \mathbb{E}\left[\frac{1}{nm}\sum_{i=1}^{n}\sum_{j=1}^{m}\left(\frac{1}{m-1}\sum_{\substack{t=1\\t\neq j}}^{m}k\left(X_{ij}, X_{it}\right) - \frac{1}{(n-1)m}\sum_{\substack{s=1\\s\neq i}}^{n}\sum_{t=1}^{m}k\left(X_{ij}, X_{st}\right)\right)\right]$$

$$= \frac{1}{nm}\sum_{i=1}^{n}\sum_{j=1}^{m}\left(\frac{1}{m-1}\sum_{\substack{t=1\\t\neq j}}^{m}\mathbb{E}\left[k\left(X_{ij}, X_{it}\right)\right] - \frac{1}{(n-1)m}\sum_{\substack{s=1\\s\neq i}}^{n}\sum_{t=1}^{m}\mathbb{E}\left[k\left(X_{ij}, X_{st}\right)\right]\right)$$

$$\stackrel{\text{Eq. 14 \& 15}}{=} \frac{1}{nm}\sum_{i=1}^{n}\sum_{j=1}^{m}\left(\frac{1}{m-1}\sum_{\substack{t=1\\t\neq j}}^{m}\mathbb{E}\left[\langle P \mid k \mid P \rangle\right] - \frac{1}{(n-1)m}\sum_{\substack{s=1\\s\neq i}}^{n}\sum_{t=1}^{m}\langle \mathbb{E}\left[P\right] \mid k \mid \mathbb{E}\left[P\right] \rangle\right)$$

$$= \mathbb{E}\left[\langle P \mid k \mid P \rangle\right] - \langle \mathbb{E}\left[P\right] \mid k \mid \mathbb{E}\left[P\right] \rangle$$
$$= \mathbb{E}\left[\langle P - \mathbb{E}\left[P\right] \mid k \mid P - \mathbb{E}\left[P\right] \rangle\right]$$
$$= \mathrm{Var}_k\left[P\right]$$

$$\tag{16}$$

### D.2.2 VARIANCE OF THE ESTIMATOR

Note that for a U-statistic $\hat{U}_n = \frac{1}{n(n-1)}\sum_{i=1}^{n}\sum_{\substack{j=1\\j\neq i}}^{n} h\left(X_i, X_j\right)$ based on i.i.d. samples $X_1, \dots, X_n$ and symmetric kernel $h$, the estimator variance is given by

$$\mathbb{V}\left(\hat{U}_n\right) = \frac{2}{n\left(n-1\right)}\mathbb{V}\left(h\left(X_1, X_2\right)\right) + \frac{4\left(n-2\right)}{n\left(n-1\right)}\mathbb{V}\left(\mathbb{E}\left[h\left(X_1, X_2\right) \mid X_2\right]\right) \tag{17}$$

(Shao, 2003).

We will use the law of total variance (TV) several times to create independence between the summands, since for dependent random variables $X$ and $Y$ with scalars $a, b$ we have $\mathbb{V}\left(aX + bY\right) = a^2\mathbb{V}\left(X\right) + b^2\mathbb{V}\left(Y\right) + 2ab\,\mathrm{Cov}\left(X, Y\right)$ (Shao, 2003).

From this also follows that

$$\mathbb{V}\left(\widehat{\mathrm{Var}}_k^{(n,m)}\right)$$

$$= \mathbb{V}\left(\frac{1}{nm}\sum_{i=1}^{n}\sum_{j=1}^{m}\left(\frac{1}{m-1}\sum_{\substack{t=1\\t\neq j}}^{m}k\left(X_{ij},X_{it}\right) - \frac{1}{(n-1)m}\sum_{\substack{s=1\\s\neq i}}^{n}\sum_{t=1}^{m}k\left(X_{ij},X_{st}\right)\right)\right)$$

$$= \mathbb{V}\left(\frac{1}{nm(m-1)}\sum_{i=1}^{n}\sum_{j=1}^{m}\sum_{\substack{t=1\\t\neq j}}^{m}k\left(X_{ij},X_{it}\right)\right) + \mathbb{V}\left(\frac{1}{n(n-1)m^2}\sum_{i=1}^{n}\sum_{j=1}^{m}\sum_{\substack{s=1\\s\neq i}}^{n}\sum_{t=1}^{m}k\left(X_{ij},X_{st}\right)\right)$$

$$- 2\,\mathrm{Cov}\left(\frac{1}{nm(m-1)}\sum_{i=1}^{n}\sum_{j=1}^{m}\sum_{\substack{t=1\\t\neq j}}^{m}k\left(X_{ij},X_{it}\right),\frac{1}{n(n-1)m^2}\sum_{i=1}^{n}\sum_{j=1}^{m}\sum_{\substack{s=1\\s\neq i}}^{n}\sum_{t=1}^{m}k\left(X_{ij},X_{st}\right)\right)$$

$$(18)$$

We will analyse each term successively and then combine the results further down in equation 27.

$$\mathbb{V}\left(\frac{1}{nm(m-1)}\sum_{i=1}^{n}\sum_{j=1}^{m}\sum_{\substack{t=1\\t\neq j}}^{m}k\left(X_{ij},X_{it}\right)\right)$$

$$\overset{\mathrm{TV}}{=} \mathbb{V}\left(\mathbb{E}\left[\frac{1}{nm(m-1)}\sum_{i=1}^{n}\sum_{j=1}^{m}\sum_{\substack{t=1\\t\neq j}}^{m}k\left(X_{ij},X_{it}\right)\mid P_1\ldots P_n\right]\right)$$

$$+ \mathbb{E}\left[\mathbb{V}\left(\frac{1}{nm(m-1)}\sum_{i=1}^{n}\sum_{j=1}^{m}\sum_{\substack{t=1\\t\neq j}}^{m}k\left(X_{ij},X_{it}\right)\mid P_1\ldots P_n\right)\right]$$

$$\overset{\mathrm{iid}}{=} \mathbb{V}\left(\frac{1}{nm(m-1)}\sum_{i=1}^{n}\sum_{j=1}^{m}\sum_{\substack{t=1\\t\neq j}}^{m}\mathbb{E}\left[k\left(X_{ij},X_{it}\right)\mid P_i\right]\right)$$

$$+ \mathbb{E}\left[\frac{1}{n^2}\sum_{i=1}^{n}\mathbb{V}\left(\frac{1}{m(m-1)}\sum_{j=1}^{m}\sum_{\substack{t=1\\t\neq j}}^{m}k\left(X_{ij},X_{it}\right)\mid P_i\right)\right]$$

$$\overset{\mathrm{iid}}{=} \mathbb{V}\left(\frac{1}{n}\sum_{i=1}^{n}\mathbb{E}\left[k\left(X_{i1},X_{i2}\right)\mid P_i\right]\right) + \frac{1}{n}\mathbb{E}\left[\mathbb{V}\left(\frac{1}{m(m-1)}\sum_{j=1}^{m}\sum_{\substack{t=1\\t\neq j}}^{m}k\left(X_{1j},X_{1t}\right)\mid P_1\right)\right]$$

$$\overset{\mathrm{iid}}{=} \frac{1}{n}\mathbb{V}\left(\langle P\mid k\mid P\rangle\right) + \frac{1}{n}\mathbb{E}\left[\mathbb{V}\left(\frac{1}{m(m-1)}\sum_{j=1}^{m}\sum_{\substack{t=1\\t\neq j}}^{m}k\left(X_{1j},X_{1t}\right)\mid P_1\right)\right]$$

$$\overset{(i)}{=} \frac{1}{n}\mathbb{V}\left(\langle P\mid k\mid P\rangle\right) + \frac{4(m-2)}{nm(m-1)}\zeta_1 + \frac{2}{nm(m-1)}\zeta_2$$

$$(19)$$

(i) with $\zeta_1 = \mathbb{E}\left[\mathbb{V}\left(\mathbb{E}\left[k\left(X_{11}, X_{12}\right) \mid X_{12}, P_1\right] \mid P_1\right)\right]$ and $\zeta_2 = \mathbb{E}\left[\mathbb{V}\left(k\left(X_{11}, X_{12}\right) \mid P_1\right)\right]$ based on equation 17.

$$
\begin{aligned}
&\mathbb{V}\left(\frac{1}{n\left(n-1\right) m^2} \sum_{i=1}^{n} \sum_{j=1}^{m} \sum_{\substack{s=1\\s\neq i}}^{n} \sum_{t=1}^{m} k\left(X_{ij}, X_{st}\right)\right) \\
&\overset{\text{TV}}{=} \underbrace{\mathbb{V}\left(\mathbb{E}\left[\frac{1}{n\left(n-1\right) m^2} \sum_{i=1}^{n} \sum_{j=1}^{m} \sum_{\substack{s=1\\s\neq i}}^{n} \sum_{t=1}^{m} k\left(X_{ij}, X_{st}\right) \mid P_1 \ldots P_n\right]\right)}_{\text{(I)}:=} \\
&+ \underbrace{\mathbb{E}\left[\mathbb{V}\left(\frac{1}{n\left(n-1\right) m^2} \sum_{i=1}^{n} \sum_{j=1}^{m} \sum_{\substack{s=1\\s\neq i}}^{n} \sum_{t=1}^{m} k\left(X_{ij}, X_{st}\right) \mid P_1 \ldots P_n\right)\right]}_{\text{(II)}:=} .
\end{aligned}
\tag{20}
$$

Due to the length of the expression, we first solve (I) and then (II).

$$
\begin{aligned}
\text{(I)} &\overset{\text{iid}}{=} \mathbb{V}\left(\frac{1}{n\left(n-1\right) m^2} \sum_{i=1}^{n} \sum_{j=1}^{m} \sum_{\substack{s=1\\s\neq i}}^{n} \sum_{t=1}^{m} \mathbb{E}\left[k\left(X_{ij}, X_{st}\right) \mid P_i, P_s\right]\right) \\
&\overset{\text{iid}}{=} \mathbb{V}\left(\frac{1}{n\left(n-1\right)} \sum_{i=1}^{n} \sum_{\substack{s=1\\s\neq i}}^{n} \langle P_i \mid k \mid P_s \rangle\right) \\
&\overset{\text{(i)}}{=} \frac{4\left(n-2\right)}{n\left(n-1\right)} \zeta_3 + \frac{2}{n\left(n-1\right)} \zeta_4
\end{aligned}
\tag{21}
$$

(i) with $\zeta_3 = \mathbb{V}\left(\mathbb{E}\left[\langle P_1 \mid k \mid P_2 \rangle \mid P_1\right]\right)$ and $\zeta_4 = \mathbb{V}\left(\langle P_1 \mid k \mid P_2 \rangle\right)$ based on equation 17.

For the next term note that $\frac{1}{m^2} \sum_{j=1}^{m} \sum_{t=1}^{m} k\left(X_{ij}, X_{st}\right)$ and $\frac{1}{m^2} \sum_{j=1}^{m} \sum_{t=1}^{m} k\left(X_{aj}, X_{bt}\right)$ are independent given $P_i, P_s, P_a, P_b$ for $i \neq s \neq a \neq b$ from which follows

$$
\text{Cov}\left(\frac{1}{m^2} \sum_{j=1}^{m} \sum_{t=1}^{m} k\left(X_{ij}, X_{st}\right), \frac{1}{m^2} \sum_{c=1}^{m} \sum_{d=1}^{m} k\left(X_{ac}, X_{bd}\right)\right) = 0.
\tag{22}
$$

Consequently, we have

$$
\begin{aligned}
\text{(II)} = \ &\mathbb{E}\left[\frac{1}{n^2(n-1)^2}\sum_{i=1}^{n}\sum_{\substack{s=1\\s\neq i}}^{n}\mathbb{V}\left(\frac{1}{m^2}\sum_{j=1}^{m}\sum_{t=1}^{m}k\left(X_{ij},X_{st}\right)\mid P_i,P_s\right)\right]\\
&+\mathbb{E}\left[\frac{1}{n^2(n-1)^2}\sum_{i=1}^{n}\sum_{\substack{s=1\\s\neq i}}^{n}\sum_{\substack{b=1\\b\neq i\\b\neq s}}^{n}\mathrm{Cov}\left(\frac{1}{m^2}\sum_{j=1}^{m}\sum_{t=1}^{m}k\left(X_{ij},X_{st}\right),\frac{1}{m^2}\sum_{c=1}^{m}\sum_{d=1}^{m}k\left(X_{ic},X_{bd}\right)\mid P_i,P_s,P_b\right)\right]\\
&+\mathbb{E}\left[\frac{1}{n^2(n-1)^2}\sum_{i=1}^{n}\sum_{\substack{s=1\\s\neq i}}^{n}\sum_{\substack{a=1\\a\neq s\\a\neq i}}^{n}\mathrm{Cov}\left(\frac{1}{m^2}\sum_{j=1}^{m}\sum_{t=1}^{m}k\left(X_{ij},X_{st}\right),\frac{1}{m^2}\sum_{c=1}^{m}\sum_{d=1}^{m}k\left(X_{ac},X_{sd}\right)\mid P_i,P_s,P_a\right)\right]\\
\stackrel{\text{sym}}{=}\ &\underbrace{\mathbb{E}\left[\frac{1}{n^2(n-1)^2}\sum_{i=1}^{n}\sum_{\substack{s=1\\s\neq i}}^{n}\mathbb{V}\left(\frac{1}{m^2}\sum_{c=1}^{m}\sum_{d=1}^{m}k\left(X_{ic},X_{sd}\right)\mid P_i,P_s\right)\right]}_{\text{(IIa)}:=}\\
&+\underbrace{\mathbb{E}\left[\frac{2}{n^2(n-1)^2}\sum_{i=1}^{n}\sum_{\substack{s=1\\s\neq i}}^{n}\sum_{\substack{b=1\\b\neq i\\b\neq s}}^{n}\mathrm{Cov}\left(\frac{1}{m^2}\sum_{j=1}^{m}\sum_{t=1}^{m}k\left(X_{ij},X_{st}\right),\frac{1}{m^2}\sum_{c=1}^{m}\sum_{d=1}^{m}k\left(X_{ic},X_{bd}\right)\mid P_i,P_s,P_b\right)\right]}_{\text{(IIb)}:=}.
\end{aligned}
\tag{23}
$$

Due to the length of the expressions, we again first look at (IIa) and then (IIb).

$$
\text{(IIa)} \stackrel{\text{iid}}{=} \mathbb{E}\left[\frac{1}{n^2\left(n-1\right)^2}\sum_{i=1}^{n}\sum_{\substack{s=1\\s\neq i}}^{n}\mathbb{V}\left(\frac{1}{m^2}\sum_{j=1}^{m}\sum_{t=1}^{m}k\left(X_{ij},X_{st}\right)\mid P_i,P_s\right)\right]
$$

$$
\stackrel{\text{iid}}{=}\frac{1}{n\left(n-1\right)}\mathbb{E}\left[\mathbb{V}\left(\frac{1}{m^2}\sum_{j=1}^{m}\sum_{t=1}^{m}k\left(X_{1j},X_{2t}\right)\mid P_1,P_2\right)\right]
$$

$$
=\frac{1}{n\left(n-1\right)}\mathbb{E}\left[\text{Cov}\left(\frac{1}{m^2}\sum_{j=1}^{m}\sum_{t=1}^{m}k\left(X_{1j},X_{2t}\right),\frac{1}{m^2}\sum_{i=1}^{m}\sum_{s=1}^{m}k\left(X_{1i},X_{2s}\right)\mid P_1,P_2\right)\right]
$$

$$
=\frac{1}{n\left(n-1\right)m^4}\sum_{j=1}^{m}\sum_{t=1}^{m}\sum_{i=1}^{m}\sum_{s=1}^{m}\mathbb{E}\left[\text{Cov}\left(k\left(X_{1j},X_{2t}\right),k\left(X_{1i},X_{2s}\right)\mid P_1,P_2\right)\right]
$$

$$
\stackrel{\text{iid}}{=}\frac{1}{n\left(n-1\right)m^4}\sum_{j=1}^{m}\sum_{t=1}^{m}\mathbb{E}\left[\text{Cov}\left(k\left(X_{1j},X_{2t}\right),k\left(X_{1j},X_{2t}\right)\mid P_1,P_2\right)\right]
$$

$$
+\frac{1}{n\left(n-1\right)m^4}\sum_{j=1}^{m}\sum_{t=1}^{m}\sum_{\substack{i=1\\i\neq j}}^{m}\mathbb{E}\left[\text{Cov}\left(k\left(X_{1j},X_{2t}\right),k\left(X_{1i},X_{2t}\right)\mid P_1,P_2\right)\right]
$$

$$
+\frac{1}{n\left(n-1\right)m^4}\sum_{j=1}^{m}\sum_{t=1}^{m}\sum_{\substack{s=1\\s\neq t}}^{m}\mathbb{E}\left[\text{Cov}\left(k\left(X_{1j},X_{2t}\right),k\left(X_{1j},X_{2s}\right)\mid P_1,P_2\right)\right]
$$

$$
\stackrel{\text{iid}}{=}\frac{1}{n\left(n-1\right)m^2}\mathbb{E}\left[\text{Cov}\left(k\left(X_{11},X_{21}\right),k\left(X_{11},X_{21}\right)\mid P_1,P_2\right)\right]
$$

$$
+\frac{m-1}{n\left(n-1\right)m^2}\mathbb{E}\left[\text{Cov}\left(k\left(X_{11},X_{21}\right),k\left(X_{12},X_{21}\right)\mid P_1,P_2\right)\right]
$$

$$
+\frac{m-1}{n\left(n-1\right)m^2}\mathbb{E}\left[\text{Cov}\left(k\left(X_{11},X_{21}\right),k\left(X_{11},X_{22}\right)\mid P_1,P_2\right)\right]
$$

$$
\stackrel{\text{(i)}}{=}\frac{1}{n\left(n-1\right)m^2}\underbrace{\mathbb{E}\left[\mathbb{V}\left(k\left(X_{11},X_{21}\right)\mid P_1,P_2\right)\right]}_{\zeta_6:=}
$$

$$
+\frac{2\left(m-1\right)}{n\left(n-1\right)m^2}\underbrace{\mathbb{E}\left[\text{Cov}\left(k\left(X_{11},X_{21}\right),k\left(X_{12},X_{21}\right)\mid P_1,P_2\right)\right]}_{\zeta_5:=}
$$

(24)

(i) follows from symmetry of $k$ and assumption of identical distributions.

Further, we have $\text{Cov}\left(k\left(X_{ij},X_{st}\right),k\left(X_{ic},X_{bd}\right)\mid P_i,P_s,P_b\right)=0$ whenever $c\neq j$ (since $s\neq b$ and independence assumption), giving

$$
\begin{aligned}
\text{(IIb)} &= \mathbb{E}\left[\frac{2}{n^2\left(n-1\right)^2}\sum_{i=1}^{n}\sum_{\substack{s=1\\s\neq i}}^{n}\sum_{\substack{b=1\\b\neq i\\b\neq s}}^{n}\text{Cov}\left(\frac{1}{m^2}\sum_{j=1}^{m}\sum_{t=1}^{m}k\left(X_{ij},X_{st}\right),\frac{1}{m^2}\sum_{c=1}^{m}\sum_{d=1}^{m}k\left(X_{ic},X_{bd}\right)\mid P_i,P_s,P_b\right)\right]\\
&= \mathbb{E}\left[\frac{2}{n^2\left(n-1\right)^2 m^4}\sum_{i=1}^{n}\sum_{\substack{s=1\\s\neq i}}^{n}\sum_{\substack{b=1\\b\neq i\\b\neq s}}^{n}\sum_{j=1}^{m}\sum_{t=1}^{m}\sum_{d=1}^{m}\text{Cov}\left(k\left(X_{ij},X_{st}\right),k\left(X_{ij},X_{bd}\right)\mid P_i,P_s,P_b\right)\right]\\
&\overset{\text{iid}}{=} \mathbb{E}\left[\frac{2}{n^2\left(n-1\right)^2 m^4}\sum_{i=1}^{n}\sum_{\substack{s=1\\s\neq i}}^{n}\sum_{\substack{b=1\\b\neq i\\b\neq s}}^{n}\sum_{j=1}^{m}\sum_{t=1}^{m}\sum_{d=1}^{m}\text{Cov}\left(k\left(X_{11},X_{21}\right),k\left(X_{11},X_{31}\right)\mid P_1,P_2,P_3\right)\right]\\
&= \frac{2\left(n-2\right)}{n\left(n-1\right)m}\underbrace{\mathbb{E}\left[\text{Cov}\left(k\left(X_{11},X_{21}\right),k\left(X_{11},X_{31}\right)\mid P_1,P_2,P_3\right)\right]}_{\zeta_9:=}
\end{aligned}
$$

$$(25)$$

The only term left is

$$
\mathrm{Cov}\left(\frac{1}{nm(m-1)}\sum_{i=1}^{n}\sum_{j=1}^{m}\sum_{\substack{t=1\\t\neq j}}^{m}k\left(X_{ij},X_{it}\right),\frac{1}{n(n-1)m^2}\sum_{i=1}^{n}\sum_{j=1}^{m}\sum_{\substack{s=1\\s\neq i}}^{n}\sum_{t=1}^{m}k\left(X_{ij},X_{st}\right)\right)
$$

$$
=\frac{1}{n^2(n-1)m^3(m-1)}\sum_{i=1}^{n}\sum_{j=1}^{m}\sum_{\substack{t=1\\t\neq j}}^{m}\sum_{o=1}^{n}\sum_{p=1}^{m}\sum_{\substack{s=1\\s\neq o}}^{n}\sum_{r=1}^{m}\mathrm{Cov}\left(k\left(X_{ij},X_{it}\right),k\left(X_{op},X_{sr}\right)\right)
$$

$$
\stackrel{(i)}{=}\frac{1}{n^2(n-1)m^3(m-1)}\sum_{i=1}^{n}\sum_{j=1}^{m}\sum_{\substack{t=1\\t\neq j}}^{m}\sum_{p=1}^{m}\sum_{\substack{s=1\\s\neq i}}^{n}\sum_{r=1}^{m}
$$
$$
\left(\mathrm{Cov}\left(k\left(X_{ij},X_{it}\right),k\left(X_{ip},X_{sr}\right)\right)+\mathrm{Cov}\left(k\left(X_{ij},X_{it}\right),k\left(X_{sp},X_{ir}\right)\right)\right)
$$

$$
\stackrel{(ii)}{=}\frac{2}{n^2(n-1)m^3(m-1)}\sum_{i=1}^{n}\sum_{j=1}^{m}\sum_{\substack{t=1\\t\neq j}}^{m}\sum_{p=1}^{m}\sum_{\substack{s=1\\s\neq i}}^{n}\sum_{r=1}^{m}\mathrm{Cov}\left(k\left(X_{ij},X_{it}\right),k\left(X_{ip},X_{sr}\right)\right)
$$

$$
\stackrel{iid}{=}\frac{2mn(n-1)}{n^2(n-1)m^3(m-1)}\sum_{j=1}^{m}\sum_{\substack{t=1\\t\neq j}}^{m}\sum_{p=1}^{m}\mathrm{Cov}\left(k\left(X_{1j},X_{1t}\right),k\left(X_{1p},X_{21}\right)\right)
$$

$$
=\frac{2}{nm^2(m-1)}\left(\sum_{j=1}^{m}\sum_{\substack{t=1\\t\neq j}}^{m}\sum_{p\in\{j,t\}}\mathrm{Cov}\left(k\left(X_{1j},X_{1t}\right),k\left(X_{1p},X_{21}\right)\right)+\sum_{j=1}^{m}\sum_{\substack{t=1\\t\neq j}}^{m}\sum_{\substack{p=1\\p\neq j\\p\neq t}}^{m}\mathrm{Cov}\left(k\left(X_{1j},X_{1t}\right),k\left(X_{1p},X_{21}\right)\right)\right)
$$

$$
\stackrel{(iii)}{=}\frac{2}{nm^2(m-1)}\left(\underbrace{\sum_{j=1}^{m}\sum_{\substack{t=1\\t\neq j}}^{m}\sum_{p\in\{j,t\}}\mathrm{Cov}\left(k\left(X_{11},X_{12}\right),k\left(X_{11},X_{21}\right)\right)}_{2m(m-1)\text{ summands}}+\underbrace{\sum_{j=1}^{m}\sum_{\substack{t=1\\t\neq j}}^{m}\sum_{\substack{p=1\\p\neq j\\p\neq t}}^{m}\mathrm{Cov}\left(k\left(X_{11},X_{12}\right),k\left(X_{13},X_{21}\right)\right)}_{m(m-1)(m-2)\text{ summands}}\right)
$$

$$
=\frac{4m(m-1)}{nm^2(m-1)}\mathrm{Cov}\left(k\left(X_{11},X_{12}\right),k\left(X_{11},X_{21}\right)\right)+\frac{2m(m-1)(m-2)}{nm^2(m-1)}\mathrm{Cov}\left(k\left(X_{11},X_{12}\right),k\left(X_{13},X_{21}\right)\right)
$$

$$
=\frac{4}{nm}\underbrace{\mathrm{Cov}\left(k\left(X_{11},X_{12}\right),k\left(X_{11},X_{21}\right)\right)}_{\zeta_7:=}+\frac{2(m-2)}{nm}\underbrace{\mathrm{Cov}\left(k\left(X_{11},X_{12}\right),k\left(X_{13},X_{21}\right)\right)}_{\zeta_8:=}.
$$

$$
(26)
$$

(i) when $i\neq o\neq s$, then $\mathrm{Cov}\left(k\left(X_{ij},X_{it}\right),k\left(X_{op},X_{sr}\right)\right)=0$.
(ii) symmetry of $k$ and iid property result in $\mathrm{Cov}\left(k\left(X_{ij},X_{it}\right),k\left(X_{ip},X_{sr}\right)\right)=\mathrm{Cov}\left(k\left(X_{ij},X_{it}\right),k\left(X_{sp},X_{ir}\right)\right)$ as long as $i\neq s$
(iii) iid and symmetry of $k$.

It follows from inserting equation 19, equation 21, equation 25, and equation 26, into equation 18 that

$$\mathbb{V}\left(\widehat{\mathrm{Var}}_k^{(n,m)}\right) = \underbrace{\frac{1}{n}\mathbb{V}\left(\langle P \mid k \mid P\rangle\right) + \frac{4(n-2)}{n(n-1)}\zeta_3 - \frac{4(m-2)}{nm}\zeta_8}_{\mathscr{O}\left(\frac{1}{n}\right)} + \underbrace{\frac{2}{n(n-1)}\zeta_4}_{\mathscr{O}\left(\frac{1}{n^2}\right)}$$

$$+ \underbrace{\frac{4(m-2)}{nm(m-1)}\zeta_1 - \frac{8}{nm}\zeta_7 + \frac{2(n-2)}{n(n-1)m}\zeta_9}_{\mathscr{O}\left(\frac{1}{nm}\right)}$$

$$+ \underbrace{\frac{2}{nm(m-1)}\zeta_2}_{\mathscr{O}\left(\frac{1}{nm^2}\right)} + \underbrace{\frac{2(m-1)}{n(n-1)m^2}\zeta_5}_{\mathscr{O}\left(\frac{1}{n^2 m}\right)} + \underbrace{\frac{1}{n(n-1)m^2}\zeta_6}_{\mathscr{O}\left(\frac{1}{n^2 m^2}\right)}.$$

$$(27)$$

In summary, our estimator is in $\mathscr{O}\left(\frac{1}{n}\left(1 + \frac{1}{m}\right)\right)$ and consequently consistent w.r.t. $n$ but not $m$.

### D.2.3 ILLUSTRATION OF THE ESTIMATOR

The estimator can also be visualized as the following. For $i, s \in \{1, \dots, n\}$ define the quadratic matrices $\mathbf{K}_{is} = \left(k_{is_{jt}}\right)_{j,t=1\dots m} \in \mathbb{R}^{m \times m}$ with entries $k_{is_{jt}} = k\left(X_{ij}, X_{st}\right)$. Then we have the colored block matrix

$$\begin{pmatrix} \mathbf{K}_{11} & \cdots & \cdots & \mathbf{K}_{n1} \\ \vdots & \ddots & \mathbf{K}_{is} & \vdots \\ \vdots & \mathbf{K}_{si} & \ddots & \vdots \\ \mathbf{K}_{1n} & \cdots & \cdots & \mathbf{K}_{nn} \end{pmatrix} = \left(\begin{array}{cccc|cc} k_{11_{11}} & \cdots & \cdots & k_{11_{1m}} & k_{12_{11}} & \cdots \\ \vdots & \ddots & k_{11_{jt}} & \vdots & \vdots & \ddots \\ \vdots & k_{11_{tj}} & \ddots & \vdots & \vdots & \ddots \\ k_{11_{m1}} & \cdots & \cdots & k_{11_{mm}} & k_{12_{m1}} & \cdots \\ \hline k_{21_{11}} & \cdots & \cdots & k_{21_{1m}} & k_{22_{11}} & \cdots \\ \vdots & \ddots & \ddots & \vdots & \vdots & \ddots \end{array}\right). \quad (28)$$

The proposed distributional variance estimator is then the average of all cyan entries without the red ones (blocks on diagonal without diagonal entries) minus the average of all black entries (off-diagonal blocks).

### D.3 DISTRIBUTIONAL COVARIANCE ESTIMATOR

Note that we have under the given i.i.d. assumptions that

$$\begin{aligned} \mathbb{E}\left[k\left(X_{ij}, Y_{it}\right)\right] &= \mathbb{E}\left[\mathbb{E}\left[k\left(X_{ij}, Y_{it}\right) \mid P_i, Q_i\right]\right] \\ &\stackrel{\text{iid}}{=} \mathbb{E}\left[\mathbb{E}\left[k\left(X_{i1}, Y_{i1}\right) \mid P_i, Q_i\right]\right] \\ &\stackrel{\text{iid}}{=} \mathbb{E}\left[\langle P_i \mid k \mid Q_i\rangle\right] \\ &\stackrel{\text{iid}}{=} \mathbb{E}\left[\langle P \mid k \mid Q\rangle\right] \end{aligned} \quad (29)$$

as well as for $i \neq s$

$$\begin{aligned} \mathbb{E}\left[k\left(X_{ij}, Y_{st}\right)\right] &= \mathbb{E}\left[\mathbb{E}\left[k\left(X_{ij}, Y_{st}\right) \mid P_i, Q_s\right]\right] \\ &\stackrel{\text{iid}}{=} \mathbb{E}\left[\mathbb{E}\left[k\left(X_{i1}, Y_{s1}\right) \mid P_i, Q_s\right]\right] \\ &\stackrel{\text{iid}}{=} \mathbb{E}\left[\langle P_i \mid k \mid Q_s\rangle\right] \\ &\stackrel{\text{iid}}{=} \langle \mathbb{E}\left[P\right] \mid k \mid \mathbb{E}\left[Q\right]\rangle. \end{aligned} \quad (30)$$

### D.3.1   EXPECTATION OF THE ESTIMATOR

Now, we can prove that the covariance estimator is unbiased, i.e.

$$
\begin{aligned}
&\mathbb{E}\left[\widehat{\operatorname{Cov}}_k^{(n,m)}(\mathbf{X},\mathbf{Y})\right] \\
&= \mathbb{E}\left[\frac{1}{nm^2}\sum_{i=1}^{n}\sum_{j=1}^{m}\sum_{t=1}^{m}\left(k\left(X_{ij},Y_{it}\right) - \frac{1}{n-1}\sum_{\substack{s=1\\s\neq i}}^{n}k\left(X_{ij},Y_{st}\right)\right)\right] \\
&= \frac{1}{nm^2}\sum_{i=1}^{n}\sum_{j=1}^{m}\sum_{t=1}^{m}\left(\mathbb{E}\left[k\left(X_{ij},Y_{it}\right)\right] - \frac{1}{n-1}\sum_{\substack{s=1\\s\neq i}}^{n}\mathbb{E}\left[k\left(X_{ij},Y_{st}\right)\right]\right) \\
&\overset{\text{Eq. 29 \& 30}}{=} \frac{1}{nm^2}\sum_{i=1}^{n}\sum_{j=1}^{m}\sum_{t=1}^{m}\left(\mathbb{E}\left[\langle P\,|\,k\,|\,Q\rangle\right] - \frac{1}{n-1}\sum_{\substack{s=1\\s\neq i}}^{n}\langle\mathbb{E}\left[P\right]\,|\,k\,|\,\mathbb{E}\left[Q\right]\rangle\right) \\
&= \mathbb{E}\left[\langle P\,|\,k\,|\,Q\rangle\right] - \langle\mathbb{E}\left[P\right]\,|\,k\,|\,\mathbb{E}\left[Q\right]\rangle \\
&= \mathbb{E}\left[\langle P - \mathbb{E}\left[P\right]\,|\,k\,|\,Q - \mathbb{E}\left[Q\right]\rangle\right] \\
&= \operatorname{Cov}_k\left(P,Q\right).
\end{aligned}
\tag{31}
$$

### D.3.2   VARIANCE OF THE ESTIMATOR

Similar to the variance case, we also analyse its convergence rate:

$$
\begin{aligned}
&\mathbb{V}\left(\widehat{\operatorname{Cov}}_k^{(n,m)}(\mathbf{X},\mathbf{Y})\right) \\
&= \mathbb{V}\left(\frac{1}{nm^2}\sum_{i=1}^{n}\sum_{j=1}^{m}\sum_{t=1}^{m}\left(k\left(X_{ij},Y_{it}\right) - \frac{1}{n-1}\sum_{\substack{s=1\\s\neq i}}^{n}k\left(X_{ij},Y_{st}\right)\right)\right) \\
&= \mathbb{V}\left(\frac{1}{nm^2}\sum_{i=1}^{n}\sum_{j=1}^{m}\sum_{t=1}^{m}k\left(X_{ij},Y_{it}\right)\right) + \mathbb{V}\left(\frac{1}{n(n-1)m^2}\sum_{i=1}^{n}\sum_{j=1}^{m}\sum_{t=1}^{m}\sum_{\substack{s=1\\s\neq i}}^{n}k\left(X_{ij},Y_{st}\right)\right) \\
&\quad - 2\operatorname{Cov}\left(\frac{1}{nm^2}\sum_{i=1}^{n}\sum_{j=1}^{m}\sum_{t=1}^{m}k\left(X_{ij},Y_{it}\right),\frac{1}{n(n-1)m^2}\sum_{i=1}^{n}\sum_{j=1}^{m}\sum_{t=1}^{m}\sum_{\substack{s=1\\s\neq i}}^{n}k\left(X_{ij},Y_{st}\right)\right)
\end{aligned}
\tag{32}
$$

$$
\mathbb{V}\left(\frac{1}{nm^2}\sum_{i=1}^{n}\sum_{j=1}^{m}\sum_{t=1}^{m}k\left(X_{ij},Y_{it}\right)\right)
$$

$$
\overset{\text{TV}}{=}\mathbb{V}\left(\mathbb{E}\left[\frac{1}{nm^2}\sum_{i=1}^{n}\sum_{j=1}^{m}\sum_{t=1}^{m}k\left(X_{ij},Y_{it}\right)\mid P_1\ldots P_n,Q_1\ldots Q_n\right]\right)
$$

$$
+\mathbb{E}\left[\mathbb{V}\left(\frac{1}{nm^2}\sum_{i=1}^{n}\sum_{j=1}^{m}\sum_{t=1}^{m}k\left(X_{ij},Y_{it}\right)\mid P_1\ldots P_n,Q_1\ldots Q_n\right)\right]
$$

$$
\overset{\text{iid}}{=}\mathbb{V}\left(\frac{1}{nm^2}\sum_{i=1}^{n}\sum_{j=1}^{m}\sum_{t=1}^{m}\mathbb{E}\left[k\left(X_{ij},Y_{it}\right)\mid P_i,Q_i\right]\right)
$$

$$
+\mathbb{E}\left[\frac{1}{n^2}\sum_{i=1}^{n}\mathbb{V}\left(\frac{1}{m^2}\sum_{j=1}^{m}\sum_{t=1}^{m}k\left(X_{ij},Y_{it}\right)\mid P_i,Q_i\right)\right] \tag{33}
$$

$$
\overset{\text{iid}}{=}\mathbb{V}\left(\frac{1}{n}\sum_{i=1}^{n}\mathbb{E}\left[k\left(X_{i1},Y_{i1}\right)\mid P_i,Q_i\right]\right)+\frac{1}{n}\mathbb{E}\left[\mathbb{V}\left(\frac{1}{m^2}\sum_{j=1}^{m}\sum_{t=1}^{m}k\left(X_{1j},Y_{1t}\right)\mid P_1,Q_1\right)\right]
$$

$$
\overset{\text{iid}}{=}\frac{1}{n}\mathbb{V}\left[\langle P\mid k\mid Q\rangle\right]+\frac{1}{n}\mathbb{E}\left[\mathbb{V}\left(\frac{1}{m^2}\sum_{j=1}^{m}\sum_{t=1}^{m}k\left(X_{1j},Y_{1t}\right)\mid P_1,Q_1\right)\right]
$$

$$
\overset{\text{iid}}{=}\frac{1}{n}\underbrace{\mathbb{V}\left(\langle P\mid k\mid Q\rangle\right)}_{\eta_1:=}+\frac{1}{nm^2}\underbrace{\mathbb{E}\left[\mathbb{V}\left(k\left(X_{11},Y_{11}\right)\mid P_1,Q_1\right)\right]}_{\eta_2:=}
$$

and for the second term we have

$$
\mathbb{V}\left(\frac{1}{n\left(n-1\right)m^2}\sum_{i=1}^{n}\sum_{j=1}^{m}\sum_{t=1}^{m}\sum_{\substack{s=1\\s\neq i}}^{n}k\left(X_{ij},Y_{st}\right)\right)
$$

$$
\overset{\text{TV}}{=}\underbrace{\mathbb{V}\left(\mathbb{E}\left[\frac{1}{n\left(n-1\right)m^2}\sum_{i=1}^{n}\sum_{j=1}^{m}\sum_{\substack{s=1\\s\neq i}}^{n}\sum_{t=1}^{m}k\left(X_{ij},Y_{st}\right)\mid P_1\ldots P_n,Q_1\ldots Q_n\right]\right)}_{\text{(III):=}} \tag{34}
$$

$$
+\underbrace{\mathbb{E}\left[\mathbb{V}\left(\frac{1}{n\left(n-1\right)m^2}\sum_{i=1}^{n}\sum_{j=1}^{m}\sum_{\substack{s=1\\s\neq i}}^{n}\sum_{t=1}^{m}k\left(X_{ij},Y_{st}\right)\mid P_1\ldots P_n,Q_1\ldots Q_n\right)\right]}_{\text{(IV):=}}.
$$

Due to the length of the expression, we first solve (III) and then (IV).

$$(\text{III}) \overset{\text{iid}}{=} \mathbb{V}\left( \frac{1}{n(n-1)m^2} \sum_{i=1}^{n} \sum_{j=1}^{m} \sum_{\substack{s=1\\s\neq i}}^{n} \sum_{t=1}^{m} \mathbb{E}\left[ k\left( X_{ij}, Y_{st} \right) \mid P_i, Q_s \right] \right)$$

$$\overset{\text{iid}}{=} \mathbb{V}\left( \frac{1}{n(n-1)m^2} \sum_{i=1}^{n} \sum_{j=1}^{m} \sum_{\substack{s=1\\s\neq i}}^{n} \sum_{t=1}^{m} \mathbb{E}\left[ k\left( X_{i1}, Y_{s1} \right) \mid P_i, Q_s \right] \right)$$

$$= \mathbb{V}\left( \frac{1}{n(n-1)} \sum_{i=1}^{n} \sum_{\substack{s=1\\s\neq i}}^{n} \langle P_i \mid k \mid Q_s \rangle \right)$$

$$= \mathbb{E}\left[ \left( \frac{1}{n(n-1)} \sum_{i=1}^{n} \sum_{\substack{s=1\\s\neq i}}^{n} \langle P_i \mid k \mid Q_s \rangle \right)^2 \right] - \left( \mathbb{E}\left[ \frac{1}{n(n-1)} \sum_{i=1}^{n} \sum_{\substack{s=1\\s\neq i}}^{n} \langle P_i \mid k \mid Q_s \rangle \right] \right)^2$$

$$\overset{\text{iid}}{=} \frac{1}{n^2(n-1)^2} \sum_{i=1}^{n} \sum_{\substack{s=1\\s\neq i}}^{n} \sum_{j=1}^{n} \sum_{\substack{t=1\\t\neq j}}^{n} \mathbb{E}\left[ \langle P_i \mid k \mid Q_s \rangle \langle P_j \mid k \mid Q_t \rangle \right] - \langle \mathbb{E}\left[ P \right] \mid k \mid \mathbb{E}\left[ Q \right] \rangle^2$$

$$= \frac{1}{n^2(n-1)^2} \sum_{i=1}^{n} \sum_{\substack{s=1\\s\neq i}}^{n} \mathbb{E}\left[ \langle P_i \mid k \mid Q_s \rangle^2 \right] + \frac{1}{n^2(n-1)^2} \sum_{i=1}^{n} \sum_{\substack{s=1\\s\neq i}}^{n} \sum_{\substack{t=1\\t\neq j\\t\neq s}}^{n} \mathbb{E}\left[ \langle P_i \mid k \mid Q_s \rangle \langle P_i \mid k \mid Q_t \rangle \right]$$

$$+ \frac{1}{n^2(n-1)^2} \sum_{i=1}^{n} \sum_{\substack{s=1\\s\neq i}}^{n} \sum_{\substack{j=1\\j\neq i\\j\neq s}}^{n} \mathbb{E}\left[ \langle P_i \mid k \mid Q_s \rangle \langle P_j \mid k \mid Q_s \rangle \right]$$

$$+ \frac{1}{n^2(n-1)^2} \sum_{i=1}^{n} \sum_{\substack{s=1\\s\neq i}}^{n} \sum_{\substack{j=1\\j\neq i\\j\neq s}}^{n} \sum_{\substack{t=1\\t\neq i\\t\neq s\\t\neq j}}^{n} \mathbb{E}\left[ \langle P_i \mid k \mid Q_s \rangle \langle P_j \mid k \mid Q_t \rangle \right] - \langle \mathbb{E}\left[ P \right] \mid k \mid \mathbb{E}\left[ Q \right] \rangle^2$$

$$\overset{\text{iid}}{=} \frac{1}{n(n-1)} \mathbb{E}\left[ \langle P_1 \mid k \mid Q_2 \rangle^2 \right] + \frac{n-2}{n(n-1)} \mathbb{E}\left[ \langle P_1 \mid k \mid Q_2 \rangle \langle P_1 \mid k \mid Q_3 \rangle \right]$$

$$+ \frac{n-2}{n(n-1)} \mathbb{E}\left[ \langle P_2 \mid k \mid Q_1 \rangle \langle P_3 \mid k \mid Q_1 \rangle \right] + \frac{(n-2)(n-3)}{n(n-1)} \langle \mathbb{E}\left[ P \right] \mid k \mid \mathbb{E}\left[ Q \right] \rangle^2 - \langle \mathbb{E}\left[ P \right] \mid k \mid \mathbb{E}\left[ Q \right] \rangle^2$$

$$= \frac{1}{n(n-1)} \mathbb{E}\left[ \langle P_1 \mid k \mid Q_2 \rangle^2 \right] + \frac{n-2}{n(n-1)} \mathbb{E}\left[ \langle P_1 \mid k \mid Q_2 \rangle \langle P_1 \mid k \mid Q_3 \rangle \right]$$

$$+ \frac{n-2}{n(n-1)} \mathbb{E}\left[ \langle P_2 \mid k \mid Q_1 \rangle \langle P_3 \mid k \mid Q_1 \rangle \right] + \frac{(n-2)(n-3) - n(n-1)}{n(n-1)} \langle \mathbb{E}\left[ P \right] \mid k \mid \mathbb{E}\left[ Q \right] \rangle^2$$

$$= \frac{1}{n(n-1)} \mathbb{E}\left[ \langle P_1 \mid k \mid Q_2 \rangle^2 \right] + \frac{n-2}{n(n-1)} \mathbb{E}\left[ \langle P_1 \mid k \mid Q_2 \rangle \langle P_1 \mid k \mid Q_3 \rangle \right]$$

$$+ \frac{n-2}{n(n-1)} \mathbb{E}\left[ \langle P_2 \mid k \mid Q_1 \rangle \langle P_3 \mid k \mid Q_1 \rangle \right] + \frac{(n-2)(n-3) - n(n-1)}{n(n-1)} \langle \mathbb{E}\left[ P \right] \mid k \mid \mathbb{E}\left[ Q \right] \rangle^2$$

$$= \frac{1}{n(n-1)} \mathbb{E}\left[ \langle P_1 \mid k \mid Q_2 \rangle^2 \right] + \frac{n-2}{n(n-1)} \mathbb{E}\left[ \langle P_1 \mid k \mid Q_2 \rangle \langle P_1 \mid k \mid Q_3 \rangle \right]$$

$$+ \frac{n-2}{n(n-1)} \mathbb{E}\left[ \langle P_2 \mid k \mid Q_1 \rangle \langle P_3 \mid k \mid Q_1 \rangle \right] - \frac{4n-6}{n(n-1)} \langle \mathbb{E}\left[ P \right] \mid k \mid \mathbb{E}\left[ Q \right] \rangle^2$$

$$\overset{\text{(i)}}{=} \frac{1}{n(n-1)} \eta_3 + \frac{n-2}{n(n-1)} \eta_4$$

$$(35)$$

(i) with $\eta_3 := \mathbb{E}\left[\langle P_1 \,|\, k \,|\, Q_2 \rangle^2\right] - 2\langle \mathbb{E}\left[P\right] \,|\, k \,|\, \mathbb{E}\left[Q\right]\rangle^2$ and $\eta_4 := \mathbb{E}\left[\langle P_1 \,|\, k \,|\, Q_2 \rangle \langle P_1 \,|\, k \,|\, Q_3 \rangle\right] + \mathbb{E}\left[\langle P_2 \,|\, k \,|\, Q_1 \rangle \langle P_3 \,|\, k \,|\, Q_1 \rangle\right] - 4\langle \mathbb{E}\left[P\right] \,|\, k \,|\, \mathbb{E}\left[Q\right]\rangle^2$.

Due to analogous reasons as in the distributional variance case, we have

$$(\mathrm{IV}) = \mathbb{E}\Bigg[\underbrace{\frac{1}{n^2(n-1)^2}\sum_{i=1}^{n}\sum_{\substack{s=1\\s\neq i}}^{n}\mathbb{V}\left(\frac{1}{m^2}\sum_{j=1}^{m}\sum_{t=1}^{m}k\left(X_{ij},Y_{st}\right)\mid P_i,Q_s\right)}_{(\mathrm{IVa}):=}\Bigg]$$

$$+ \mathbb{E}\Bigg[\underbrace{\frac{1}{n^2(n-1)^2}\sum_{i=1}^{n}\sum_{\substack{s=1\\s\neq i}}^{n}\sum_{\substack{b=1\\b\neq i\\b\neq s}}^{n}\mathrm{Cov}\left(\frac{1}{m^2}\sum_{j=1}^{m}\sum_{t=1}^{m}k\left(X_{ij},Y_{st}\right),\frac{1}{m^2}\sum_{c=1}^{m}\sum_{d=1}^{m}k\left(X_{ic},Y_{bd}\right)\mid P_i,Q_b,Q_s\right)}_{(\mathrm{IVb}):=}\Bigg]$$

$$+ \mathbb{E}\Bigg[\underbrace{\frac{1}{n^2(n-1)^2}\sum_{i=1}^{n}\sum_{\substack{s=1\\s\neq i}}^{n}\sum_{\substack{a=1\\a\neq s\\a\neq i}}^{n}\mathrm{Cov}\left(\frac{1}{m^2}\sum_{j=1}^{m}\sum_{t=1}^{m}k\left(X_{ij},Y_{st}\right),\frac{1}{m^2}\sum_{c=1}^{m}\sum_{d=1}^{m}k\left(X_{ac},Y_{sd}\right)\mid P_i,P_a,Q_s\right)}_{(\mathrm{IVc}):=}\Bigg].$$

$$(36)$$

Due to the length of the expressions, we again first look at (IVa), then (IVb), and then (IVc).

In the following equation, we use almost the identical steps as in Equation 24:

$$
\begin{aligned}
\text{(IVa)} &= \mathbb{E}\left[\frac{1}{n^2(n-1)^2}\sum_{i=1}^{n}\sum_{\substack{s=1\\s\neq i}}^{n}\mathbb{V}\left(\frac{1}{m^2}\sum_{j=1}^{m}\sum_{t=1}^{m}k\left(X_{ij},Y_{st}\right)\mid P_i,Q_s\right)\right]\\
&\overset{\text{iid}}{=}\frac{1}{n(n-1)}\mathbb{E}\left[\mathbb{V}\left(\frac{1}{m^2}\sum_{j=1}^{m}\sum_{t=1}^{m}k\left(X_{1j},Y_{2t}\right)\mid P_1,Q_2\right)\right]\\
&=\frac{1}{n(n-1)}\mathbb{E}\left[\mathrm{Cov}\left(\frac{1}{m^2}\sum_{j=1}^{m}\sum_{t=1}^{m}k\left(X_{1j},Y_{2t}\right),\frac{1}{m^2}\sum_{i=1}^{m}\sum_{s=1}^{m}k\left(X_{1i},Y_{2s}\right)\mid P_1,Q_2\right)\right]\\
&=\frac{1}{n(n-1)m^4}\sum_{j=1}^{m}\sum_{t=1}^{m}\sum_{i=1}^{m}\sum_{s=1}^{m}\mathbb{E}\left[\mathrm{Cov}\left(k\left(X_{1j},Y_{2t}\right),k\left(X_{1i},Y_{2s}\right)\mid P_1,Q_2\right)\right]\\
&\overset{\text{iid}}{=}\frac{1}{n(n-1)m^4}\sum_{j=1}^{m}\sum_{t=1}^{m}\mathbb{E}\left[\mathrm{Cov}\left(k\left(X_{1j},Y_{2t}\right),k\left(X_{1j},Y_{2t}\right)\mid P_1,Q_2\right)\right]\\
&\quad+\frac{1}{n(n-1)m^4}\sum_{j=1}^{m}\sum_{t=1}^{m}\sum_{\substack{i=1\\i\neq j}}^{m}\mathbb{E}\left[\mathrm{Cov}\left(k\left(X_{1j},Y_{2t}\right),k\left(X_{1i},Y_{2t}\right)\mid P_1,Q_2\right)\right]\\
&\quad+\frac{1}{n(n-1)m^4}\sum_{j=1}^{m}\sum_{t=1}^{m}\sum_{\substack{s=1\\s\neq t}}^{m}\mathbb{E}\left[\mathrm{Cov}\left(k\left(X_{1j},Y_{2t}\right),k\left(X_{1j},Y_{2s}\right)\mid P_1,Q_2\right)\right]\\
&\overset{\text{iid}}{=}\frac{1}{n(n-1)m^2}\underbrace{\mathbb{E}\left[\mathrm{Cov}\left(k\left(X_{11},Y_{21}\right),k\left(X_{11},Y_{21}\right)\mid P_1,Q_2\right)\right]}_{\eta_5:=}\\
&\quad+\frac{m-1}{n(n-1)m^2}\underbrace{\mathbb{E}\left[\mathrm{Cov}\left(k\left(X_{11},Y_{21}\right),k\left(X_{12},Y_{21}\right)\mid P_1,Q_2\right)\right]}_{\eta_6:=}\\
&\quad+\frac{m-1}{n(n-1)m^2}\underbrace{\mathbb{E}\left[\mathrm{Cov}\left(k\left(X_{11},Y_{21}\right),k\left(X_{11},Y_{22}\right)\mid P_1,Q_2\right)\right]}_{\eta_7:=}.
\end{aligned}
$$

$$(37)$$

Next, we have analogous to Equation 25

$$
\begin{aligned}
(\text{IVb}) &= \mathbb{E}\left[\frac{1}{n^2(n-1)^2}\sum_{i=1}^{n}\sum_{\substack{s=1\\s\neq i}}^{n}\sum_{\substack{b=1\\b\neq i\\b\neq s}}^{n}\mathrm{Cov}\left(\frac{1}{m^2}\sum_{j=1}^{m}\sum_{t=1}^{m}k\left(X_{ij},Y_{st}\right),\frac{1}{m^2}\sum_{c=1}^{m}\sum_{d=1}^{m}k\left(X_{ic},Y_{bd}\right)\mid P_i,Q_s,Q_b\right)\right]\\
&= \mathbb{E}\left[\frac{1}{n^2(n-1)^2 m^4}\sum_{i=1}^{n}\sum_{\substack{s=1\\s\neq i}}^{n}\sum_{\substack{b=1\\b\neq i\\b\neq s}}^{n}\sum_{j=1}^{m}\sum_{t=1}^{m}\sum_{d=1}^{m}\mathrm{Cov}\left(k\left(X_{ij},Y_{st}\right),k\left(X_{ij},Y_{bd}\right)\mid P_i,Q_s,Q_b\right)\right]\\
&\overset{\text{iid}}{=} \mathbb{E}\left[\frac{1}{n^2(n-1)^2 m^4}\sum_{i=1}^{n}\sum_{\substack{s=1\\s\neq i}}^{n}\sum_{\substack{b=1\\b\neq i\\b\neq s}}^{n}\sum_{j=1}^{m}\sum_{t=1}^{m}\sum_{d=1}^{m}\mathrm{Cov}\left(k\left(X_{11},Y_{21}\right),k\left(X_{11},Y_{31}\right)\mid P_1,Q_2,Q_3\right)\right]\\
&= \frac{n-2}{n(n-1)m}\underbrace{\mathbb{E}\left[\mathrm{Cov}\left(k\left(X_{11},Y_{21}\right),k\left(X_{11},Y_{31}\right)\mid P_1,Q_2,Q_3\right)\right]}_{\eta_9:=}
\end{aligned}
$$

$$(38)$$

and in an almost identical manner

$$
\begin{aligned}
(\text{IVc}) &= \mathbb{E}\left[\frac{1}{n^2(n-1)^2}\sum_{i=1}^{n}\sum_{\substack{s=1\\s\neq i}}^{n}\sum_{\substack{a=1\\a\neq s\\a\neq i}}^{n}\mathrm{Cov}\left(\frac{1}{m^2}\sum_{j=1}^{m}\sum_{t=1}^{m}k\left(X_{ij},Y_{st}\right),\frac{1}{m^2}\sum_{c=1}^{m}\sum_{d=1}^{m}k\left(X_{ac},Y_{sd}\right)\mid P_i,P_a,Q_s\right)\right]\\
&= \mathbb{E}\left[\frac{1}{n^2(n-1)^2 m^4}\sum_{i=1}^{n}\sum_{\substack{s=1\\s\neq i}}^{n}\sum_{\substack{b=1\\b\neq i\\b\neq s}}^{n}\sum_{j=1}^{m}\sum_{t=1}^{m}\sum_{d=1}^{m}\mathrm{Cov}\left(k\left(X_{ij},Y_{st}\right),k\left(X_{ac},Y_{sd}\right)\mid P_i,P_a,Q_s\right)\right]\\
&\overset{\text{iid}}{=} \mathbb{E}\left[\frac{1}{n^2(n-1)^2 m^4}\sum_{i=1}^{n}\sum_{\substack{s=1\\s\neq i}}^{n}\sum_{\substack{b=1\\b\neq i\\b\neq s}}^{n}\sum_{j=1}^{m}\sum_{t=1}^{m}\sum_{d=1}^{m}\mathrm{Cov}\left(k\left(X_{11},Y_{31}\right),k\left(X_{21},Y_{31}\right)\mid P_1,P_2,Q_3\right)\right]\\
&= \frac{n-2}{n(n-1)m}\underbrace{\mathbb{E}\left[\mathrm{Cov}\left(k\left(X_{11},Y_{31}\right),k\left(X_{21},Y_{31}\right)\mid P_1,P_2,Q_3\right)\right]}_{\eta_{10}:=}.
\end{aligned}
$$

$$(39)$$

The only term left is

$$
\text{Cov}\left(\frac{1}{nm^2}\sum_{i=1}^{n}\sum_{j=1}^{m}\sum_{t=1}^{m}k\left(X_{ij},Y_{it}\right),\frac{1}{n\left(n-1\right)m^2}\sum_{i=1}^{n}\sum_{j=1}^{m}\sum_{t=1}^{m}\sum_{\substack{s=1\\s\neq i}}^{n}k\left(X_{ij},Y_{st}\right)\right)
$$

$$
=\frac{1}{n^2\left(n-1\right)m^4}\sum_{i=1}^{n}\sum_{j=1}^{m}\sum_{t=1}^{m}\sum_{o=1}^{n}\sum_{p=1}^{m}\sum_{\substack{s=1\\s\neq o}}^{n}\sum_{r=1}^{m}\text{Cov}\left(k\left(X_{ij},Y_{it}\right),k\left(X_{op},Y_{sr}\right)\right)
$$

$$
\stackrel{(i)}{=}\frac{1}{n^2\left(n-1\right)m^4}\sum_{i=1}^{n}\sum_{j=1}^{m}\sum_{\substack{t=1\\t\neq j}}^{m}\sum_{p=1}^{m}\sum_{\substack{s=1\\s\neq i}}^{n}\sum_{r=1}^{m}\text{Cov}\left(k\left(X_{ij},Y_{it}\right),k\left(X_{ip},Y_{sr}\right)\right)
$$

$$
+\frac{1}{n^2\left(n-1\right)m^4}\sum_{i=1}^{n}\sum_{j=1}^{m}\sum_{\substack{t=1\\t\neq j}}^{m}\sum_{p=1}^{m}\sum_{\substack{s=1\\s\neq i}}^{n}\sum_{r=1}^{m}\text{Cov}\left(k\left(X_{ij},Y_{it}\right),k\left(X_{sp},Y_{ir}\right)\right)
$$

$$
\stackrel{\text{iid}}{=}\frac{1}{nm^3}\sum_{j=1}^{m}\sum_{\substack{t=1\\t\neq j}}^{m}\sum_{p=1}^{m}\text{Cov}\left(k\left(X_{1j},Y_{1t}\right),k\left(X_{1p},Y_{21}\right)\right)+\frac{1}{nm^3}\sum_{j=1}^{m}\sum_{\substack{t=1\\t\neq j}}^{m}\sum_{r=1}^{m}\text{Cov}\left(k\left(X_{1j},Y_{1t}\right),k\left(X_{21},Y_{1r}\right)\right)
$$

$$
=\frac{1}{nm^3}\sum_{j=1}^{m}\sum_{\substack{t=1\\t\neq j}}^{m}\text{Cov}\left(k\left(X_{1j},Y_{1t}\right),k\left(X_{1j},Y_{21}\right)\right)+\frac{1}{nm^3}\sum_{j=1}^{m}\sum_{\substack{t=1\\t\neq j}}^{m}\text{Cov}\left(k\left(X_{1j},Y_{1t}\right),k\left(X_{1t},Y_{21}\right)\right)
$$

$$
+\frac{1}{nm^3}\sum_{j=1}^{m}\sum_{\substack{t=1\\t\neq j}}^{m}\sum_{\substack{p=1\\p\neq j\\p\neq t}}^{m}\text{Cov}\left(k\left(X_{1j},Y_{1t}\right),k\left(X_{1p},Y_{21}\right)\right)+\frac{1}{nm^3}\sum_{j=1}^{m}\sum_{\substack{t=1\\t\neq j}}^{m}\text{Cov}\left(k\left(X_{1j},Y_{1t}\right),k\left(X_{21},Y_{1j}\right)\right)
$$

$$
+\frac{1}{nm^3}\sum_{j=1}^{m}\sum_{\substack{t=1\\t\neq j}}^{m}\text{Cov}\left(k\left(X_{1j},Y_{1t}\right),k\left(X_{21},Y_{1t}\right)\right)+\frac{1}{nm^3}\sum_{j=1}^{m}\sum_{\substack{t=1\\t\neq j}}^{m}\sum_{\substack{p=1\\p\neq j\\p\neq t}}^{m}\text{Cov}\left(k\left(X_{1j},Y_{1t}\right),k\left(X_{21},Y_{1p}\right)\right)
$$

$$
\stackrel{\text{iid}}{=}\frac{1}{nm^3}\sum_{j=1}^{m}\sum_{\substack{t=1\\t\neq j}}^{m}\text{Cov}\left(k\left(X_{11},Y_{12}\right),k\left(X_{11},Y_{21}\right)\right)+\frac{1}{nm^3}\sum_{j=1}^{m}\sum_{\substack{t=1\\t\neq j}}^{m}\text{Cov}\left(k\left(X_{11},Y_{12}\right),k\left(X_{12},Y_{21}\right)\right)
$$

$$
+\frac{1}{nm^3}\sum_{j=1}^{m}\sum_{\substack{t=1\\t\neq j}}^{m}\sum_{\substack{p=1\\p\neq j\\p\neq t}}^{m}\text{Cov}\left(k\left(X_{11},Y_{12}\right),k\left(X_{13},Y_{21}\right)\right)+\frac{1}{nm^3}\sum_{j=1}^{m}\sum_{\substack{t=1\\t\neq j}}^{m}\text{Cov}\left(k\left(X_{11},Y_{12}\right),k\left(X_{21},Y_{11}\right)\right)
$$

$$
+\frac{1}{nm^3}\sum_{j=1}^{m}\sum_{\substack{t=1\\t\neq j}}^{m}\text{Cov}\left(k\left(X_{11},Y_{12}\right),k\left(X_{21},Y_{12}\right)\right)+\frac{1}{nm^3}\sum_{j=1}^{m}\sum_{\substack{t=1\\t\neq j}}^{m}\sum_{\substack{p=1\\p\neq j\\p\neq t}}^{m}\text{Cov}\left(k\left(X_{11},Y_{12}\right),k\left(X_{21},Y_{13}\right)\right)
$$

$$
=\frac{m-1}{nm^2}\underbrace{\left(\text{Cov}\left(k\left(X_{11},Y_{12}\right),k\left(X_{11},Y_{21}\right)\right)+\text{Cov}\left(k\left(X_{11},Y_{12}\right),k\left(X_{21},Y_{11}\right)\right)\right)}_{\eta_{11}:=}
$$

$$
+\frac{m-1}{nm^2}\underbrace{\left(\text{Cov}\left(k\left(X_{11},Y_{12}\right),k\left(X_{21},Y_{12}\right)\right)+\text{Cov}\left(k\left(X_{11},Y_{12}\right),k\left(X_{12},Y_{21}\right)\right)\right)}_{\eta_{12}:=}
$$

$$
+\frac{\left(m-1\right)\left(m-2\right)}{nm^2}\underbrace{\left(\text{Cov}\left(k\left(X_{11},Y_{12}\right),k\left(X_{13},Y_{21}\right)\right)+\text{Cov}\left(k\left(X_{11},Y_{12}\right),k\left(X_{21},Y_{13}\right)\right)\right)}_{\eta_{13}:=}.
$$

$$(40)$$

By combining all previous equations, we get

$$
\mathbb{V}\left(\widehat{\mathrm{Cov}}_k^{(n,m)}(\mathbf{X}, \mathbf{Y})\right)
$$

$$
= \underbrace{\frac{1}{n}\eta_1 + \frac{n-2}{n(n-1)}\eta_4 + \frac{-2(m-1)(m-2)}{nm^2}\eta_{13}}_{\mathscr{O}\left(\frac{1}{n}\right)} + \underbrace{\frac{1}{n(n-1)}\eta_3}_{\mathscr{O}\left(\frac{1}{n^2}\right)}
$$

$$
+ \underbrace{\frac{-2(m-1)}{nm^2}(\eta_{11} + \eta_{12}) + \frac{n-2}{n(n-1)m}(\eta_9 + \eta_{10})}_{\mathscr{O}\left(\frac{1}{nm}\right)} \tag{41}
$$

$$
+ \underbrace{\frac{1}{nm^2}\eta_2}_{\mathscr{O}\left(\frac{1}{nm^2}\right)} + \underbrace{\frac{m-1}{n(n-1)m^2}(\eta_6 + \eta_7)}_{\mathscr{O}\left(\frac{1}{n^2 m}\right)} + \underbrace{\frac{1}{n(n-1)m^2}\eta_5}_{\mathscr{O}\left(\frac{1}{n^2 m^2}\right)}.
$$

