# OpenReview forum: "A Bias-Variance-Covariance Decomposition of Kernel Scores for Generative Models"
_ICLR.cc/2024/Conference — Submitted to ICLR 2024_

### Official Review · Reviewer_ox7H · 2023-10-24

**Soundness:** 3 good
**Presentation:** 2 fair
**Contribution:** 3 good
**Rating:** 6
**Confidence:** 2

**Summary:**

This paper focuses on the evaluation of generative models’ performance. Generalization and uncertainty are two primary consideration and the bias-variance-covariance decomposition are introduced for kernel scores and entropy. The performance is examined on vision, voice and text datasets.

**Strengths:**

This paper is clearly written and the studied topic is crucial in large language models.
The author develops the distributional correlation as a tool to insight into the generative model’s fitting.
The author performed extensive experiments to validate the effectiveness of the developed tool.

**Weaknesses:**

See the question part below.

**Questions:**

Can you explain the reason why this method works best on question answering datasets?
I’m wondering how to generalize the theory to others except the kernel scores.

---

> ### Author Response · Authors · 2023-11-17
> **Overview**
>
> Dear reviewer ox7H,
>
> Thank you very much for taking the time to engage with our paper. Below, we have tried to address all of your feedback and questions.
> Specifically, we give reasoning for our strong performance in natural language generation and explain why a covariance term cannot be derived for other losses.
> Please take a look and let us know in case you would like additional clarification on any of these points.

---

> ### Author Response · Authors · 2023-11-17
> **Reasoning for the Natural Language Performance**
>
> Kernel scores are loss functions giving good values when the prediction is close to the ground truth and bad values when not. In Figure 4 and 6, the Pearson correlation between the kernel entropy (our uncertainty measure) and the loss (kernel score) shows that kernel entropy is very effective in predicting the loss.
> The nlg results in Figure 7 confirm that this holds for the loss used to evaluate question answering datasets.
> Part of the explanation for our good results is that the embedder transforms the generated answers into a semantically meaningful vector space, in which the kernel then compares the similarities. Vector embeddings combined with the cosine similarity have a long history in the NLP domain, which underlines their effectiveness (Camacho-Collados et al, 2018).
>
> **References:**
>
> Jose Camacho-Collados and Mohammad Taher Pilehvar. From word to sense embeddings: A survey on vector representations of meaning. Journal of Artificial Intelligence Research, 63:743–788, 2018.

---

> ### Author Response · Authors · 2023-11-17
> **Generalizing Theory to other Losses**
>
> We were able to derive the bias-variance-covariance decomposition by using a quadratic form in the definition of kernel scores. This quadratic form is in general not present for other losses, which prohibits deriving a covariance term.

---

### Official Review · Reviewer_27VK · 2023-10-24

**Soundness:** 3 good
**Presentation:** 3 good
**Contribution:** 3 good
**Rating:** 6
**Confidence:** 2

**Summary:**

This paper presents a method to analyze the performance of a generative model in terms of bias (compared to the true distribution) vs. variance (with respect to replications of the model on independent training data), using kernel methods to estimate distributional distances.  It develops theory which defines RKHS-based variance scores and develops simple U-statistic-like estimators.  They demonstrate their approach in image generation (Infinite MNIST), Audio Generation (LJSpeech), and Natural Language Generation examples.  In image generation example, the method shows that the variance stays high and is reduced throughout training, while bias quickly converges.

**Strengths:**

This paper contributes a useful theoretical tool.  it is clearly written.  The image generation and audio generation examples demonstrate that the method can provide useful insights into the properties (e.g. distributional diversity, stability) of the generative models.  For example, the paper demonstrates some suggestive evidence for how mode collapse affects only the bias but not the variance.  We see that using predictive kernel entropy of LLM answers to predict answer quality can be a useful tool, and compares favorably to lexical similarity and semantic entropy.

**Weaknesses:**

Practitioners may find it challenging to apply this method to their case.  It is not clear what considerations should guide selection of a kernel.
Some details about the experiments are missing, which make it harder to interpret the results (see Questions).

**Questions:**

1. In figure 4, what are the points being plotted?  The 20 different models?
2. Why is only a single model used to compute kernel entropy? Are there any the advantages of using multiple models to compute the entropy?
3. How is the bias calculated in the mode-collapse experiment of section 5.1?  Is the original distribution used to compute the bias, or the modified distribution with reduced frequency of class 0 used?
4. Why use RBF kernel for images, but Laplace kernel for audio?
5. Why do we see an entropy increase as training progresses for audio, but not for images?  Could it be an artifact of the choice to use a different kernel for images vs audio?
6. In the language model experiment, does the superiority of using kernel entropy for predicting correct answers depend on the choice of kernel?  How are the AUC scores for Laplace kernel?

---

> ### Author Response · Authors · 2023-11-17
> **Overview**
>
> Dear reviewer 27VK,
>
> We appreciate your thorough review of our paper. In response to your feedback and questions, we have made efforts to address them comprehensively below. Specifically, we update Appendix C.2 with evaluations of the RBF kernel for the audio experiments and the Laplacian and polynomial kernel for the language experiments. We discuss the results in the following and also explain why kernel entropy stays constant in the image experiments but increases in the audio experiments.
> Last, we give answers to the other miscellaneous questions.
> If you require further clarification on any specific points, please feel free to let us know.

---

> > ### Comment · Reviewer_27VK · 2023-11-21
> > **Thanks for your replies**
> >
> > Thanks for addressing my questions.  I still need more time to digest the new additions, but I will consider raising my score during the reviewer-reviewer discussion period.

---

> ### Author Response · Authors · 2023-11-17
> **RBF Kernel for Audio Experiments**
>
> The RBF kernel is a common kernel for low-to-medium dimensional data, such as low resolution images like MNIST (Schölkopf, 1997).
> However, the RBF kernel does not scale well to higher dimensions (Binkowski et al, 2018).
> This is problematic since audio instances are represented by vectors of length 100,000 or more in our experiments (c.f. Figure 10 Appendix C.2.2).
> Consequently, we use the Laplacian kernel in the main paper but also add an evaluation with the RBF kernel in Figure 11 in Appendix C.2.2.
> As expected, the results are more erratic compared to the Laplacian kernel but show a similar trend.
> Specifically, the kernel score and kernel entropy increase and decrease at the same training iterations, however the absolute Pearson correlation between them is only between 0.8 and 0.9.
>
> **References:**
>
> Bernhard Schölkopf. Support vector learning. PhD thesis, Oldenbourg München, Germany, 1997.
>
> Mikołaj Bínkowski, Danica J Sutherland, Michael Arbel, and Arthur Gretton. Demystifying MMD GANs. In International Conference on Learning Representations, 2018.

---

> ### Author Response · Authors · 2023-11-17
> **Natural Language Results for Different Kernels**
>
> We compare the AUROC of the kernel entropy for various choices of kernels in Figure 14 in Appendix C.2.3.
> The differences between RBF and polynomial kernel as well as cosine similarity are marginal, while the Laplacian kernel performs worse.
> This is in line with common practice in NLP, where cosine similarity is the most often used kernel to compare text embeddings (Camacho-Collados & Pilehvar, 2018).
>
> **References:**
>
> Jose Camacho-Collados and Mohammad Taher Pilehvar. From word to sense embeddings: A survey on vector representations of meaning. Journal of Artificial Intelligence Research, 63:743–788, 2018.

---

> ### Author Response · Authors · 2023-11-17
> **Kernel Entropy Stays Constant in Images but Increases in Audio**
>
> The discrepancy of the kernel entropy behavior between the image and audio experiments can be explained via the architecture and task setup. For image generation, the initial diffusion model already generates volatile and noisy samples. This leads to a roughly constant kernel entropy throughout training. In contrast, the audio model initially generates very short audio instances (c.f. Figure 10 Appendix C.2.2).
> This means that most of the entries in the audio vector are zeros, which results in a small kernel entropy. Consequently, the kernel entropy increases as soon as the model learns to produce longer audio instances.

---

> ### Author Response · Authors · 2023-11-17
> **Miscellaneous Questions**
>
> * Q: “Practitioners may find it challenging to apply this method to their case.”
>
> A: We added a description of our procedure to compute the kernel entropy in Algorithm 1 Appendix C.1. Note that our final approach is simpler to implement than semantic entropy introduced by Kuhn et al (2023).
>
>
> * Q: “In figure 4, what are the points being plotted? The 20 different models?”
>
> A: Each point in Figure 4 corresponds to a digit.
>
>
> * Q: “Why is only a single model used to compute kernel entropy? Are there any advantages of using multiple models to compute the entropy?”
>
> A: We compute the kernel entropy based on single models to stay as truthful as possible to practical constraints: Generative models are usually computationally very expensive and closed-source access is almost always provided only to single models. It is likely that the average kernel entropy of multiple models gives even better results because this improves the estimation.
>
>
> * Q: “How is the bias calculated in the mode-collapse experiment of section 5.1? Is the original distribution used to compute the bias, or the modified distribution with reduced frequency of class 0 used?”
>
> A: The MMD, bias, and variance values in Figure 5 are only computed with respect to digit ‘0’ (which has reduced frequency in the training data).
> The other digits in the mode-collapse experiment have similar lines as in Figure 3 (no reduced frequency in the training data).
>
>
> **References:**
>
> Lorenz Kuhn, Yarin Gal, and Sebastian Farquhar. Semantic uncertainty: Linguistic invariances for uncertainty estimation in natural language generation. In The Eleventh International Conference on Learning Representations, 2023.

---

### Official Review · Reviewer_vzDu · 2023-10-31

**Soundness:** 3 good
**Presentation:** 2 fair
**Contribution:** 2 fair
**Rating:** 6
**Confidence:** 2

**Summary:**

In this paper, the authors propose the bias-variance-covariance decomposition for kernel scores, which is an unbiased and consistent estimator for uncertainty measurement in deep generative models such as large language models. Their approach only requires samples from the predictive distribution rather than the distributions themselves, which means that we can evaluate all terms in the composition for any deep generative model.
Empirically they evaluate their approach on models with image and video data and large language models, and they showed that their kernel  entropy outperforms other baselines.

**Strengths:**

This paper is well-motivated; It is worth studying the generalization and uncertainty characterization in large language models. The paper is clearly written and well structured. To my knowledge, the math derivation in the methodology is sound. I found it useful that their estimator only would require samples from the predictive distributions rather than those distributions themselves, which has a large range of applications. Also, they provide comprehensive evaluations of both image data and text data.

**Weaknesses:**

The authors claim that one of the advantages of their uncertainty estimator is that it includes a covariance term. I would hope to see a more detailed discussion on this. For example, how that covariance term characterizes uncertainty that cannot be done by prior work.

**Questions:**

1. How do different sample sizes affect the empirical results? In Figure.2 there is a fixed sample size. I wonder how to tune that in general.
2. Is there any difference in terms of the outperformance of your estimator when you work on image data and text data?

---

> ### Author Response · Authors · 2023-11-17
> **Overview**
>
> Dear reviewer vzDu,
>
> We appreciate your thorough review of our paper. We have carefully addressed all the feedback and queries you provided below. Specifically, we discuss the effect of different sample sizes and the benefits of an explicit covariance term in the bias-variance decomposition.
> Feel free to reach out if you need further clarification on any of these matters.

---

> ### Author Response · Authors · 2023-11-17
> **Effect of Sample Sizes**
>
> In general, our theoretical results and Figure 2 indicate that more samples are always better. However, at some point the improvements are too marginal to justify the computational costs. To highlight this point, we plot the AUROC of the kernel entropy in case of Opt-13b and CoQA for different sample sizes (=number of text generations) in Figure 15 in Appendix C.2.3. As can be seen, more samples are always better, but the improvement flattens for larger sizes. In the main paper in Figure 7, we use 20 generations for CoQA and 10 generations for TriviaQA, which follows Kuhn et al (2023).
>
> **References:**
>
> Lorenz Kuhn, Yarin Gal, and Sebastian Farquhar. Semantic uncertainty: Linguistic invariances for uncertainty estimation in natural language generation. In The Eleventh International Conference on Learning Representations, 2023.

---

> ### Author Response · Authors · 2023-11-17
> **Benefits of the Covariance Term**
>
> In previous work, the bias-variance decomposition is often used to motivate ensemble approaches. Specifically, it is assumed that the individual ensemble members are independent, which then reduces the variance term and also the generalization error. However, it is not clear how negative the effect on the generalization error is when this assumption is violated.
> Our variance-covariance decomposition allows us to quantify the negative impact of correlated ensemble members via the covariance term. In Figure 3 we use the normalized covariances (=correlations) for better interpretability. As we can see, the correlations between epochs 20 to 40 are very high, which means that creating an ensemble based on iterative gradient steps, like stochastic weight averaging Gaussian (Maddox et al, 2019), strongly violates the independence assumption. Specifically, we can now quantify how much this correlation/covariance impacts the generalization error. This underlines the practical relevance of our theory.
>
> **References:**
>
> Maddox, W. J., Izmailov, P., Garipov, T., Vetrov, D. P., & Wilson, A. G. (2019). A simple baseline for bayesian uncertainty in deep learning. Advances in neural information processing systems, 32.

---

### Official Review · Reviewer_XNge · 2023-11-02

**Soundness:** 2 fair
**Presentation:** 2 fair
**Contribution:** 2 fair
**Rating:** 5
**Confidence:** 1

**Summary:**

The authors propose a bias-variance-covariance decomposition for kernel scores, applying this decomposition to generative models. In particular they aim at assessing uncertainty of predictions by looking at the derived variance term (and the related predictive kernel entropy) as measures of uncertainty.

**Strengths:**

- The work is well-motivated, since evaluating predictive uncertainty for generative models is a relevant problem.
- The writing of the paper is clear and easy to follow.
- The results in Section 5.3 indicate that the proposed approach favourably compares to recent relevant work on assessing predictive uncertainty of LLMs in QA tasks.

**Weaknesses:**

Since the paper falls far from my area of expertise, and I am not sufficiently familiar with the mentioned related work, I'll refrain from commenting on the validity of the theoretical/methodological contribution.  However, specifically to the empirical results in the paper:
- A large part of the empirical results (aside from Section 5.3) do not seem to contain important and conclusive insights. The main take-away seem to be distributional variance showing correlation with MMD, while many other observations such as on the stability of training do not seem to be as valuable.
- "This includes the discovery that mode collapse of underrepresented minority groups is expressed purely in the bias." made as a claim in the Introduction should be probably revisited. As far as I understand, the authors conclude this from their results on a single model and a single experiment, which is not enough to make a general statement.
- While the paper is largely motivated by the need for evaluating predictive uncertainty for generative models, the only assessment of the approach for this purpose is made for LLMs on QA task (Section 5.3). The authors claim that their proposed uncertainty measure is applicable to a large range of generative models, and that this is an advantage over previously proposed methods, so it would be important to test its effectiveness in assessing predictive uncertainty for other generative tasks (e.g. image generation).

**Questions:**

To fully understand the setup and results in Section 5.3 I had to open the work from which the setup is replicated [1]. I think the section would benefit from a more thorough introduction of the experiment, in order to be self-contained.

[1] Lorenz Kuhn, Yarin Gal, and Sebastian Farquhar. Semantic uncertainty: Linguistic invariances for uncertainty estimation in natural language generation. In The Eleventh International Conference on Learning Representations, 2023.

---

> ### Author Response · Authors · 2023-11-17
> **Overview**
>
> Dear reviewer XNge,
>
> Thank you for your time and attention. We appreciate your thorough review of our paper. We have carefully considered all your feedback and questions in the following responses. Specifically, we communicate how we adjust our claims about mode collapse and how image, audio, and language experiments are connected with uncertainty estimation. If you need further clarification on any of these points, please feel free to let us know.

---

> ### Author Response · Authors · 2023-11-17
> **Adjusting Claims about Mode Collapse**
>
> We agree with the reviewer that the provided evidence is not sufficient to back our strong claims about mode collapse. We now soften our claims/formulations and clarify that our bias-variance-covariance decomposition can be used as a tool to diagnose mode collapse or overfitting and elucidate its nature in terms of bias and variance. We now also provide additional experiments to showcase the practical utility and repeat the same mode-collapse experiment of Figure 5 with other digits reduced (digit ‘2’ and ‘3’, c.f. Figure 9 Appendix C.2.1). Based on our bias-variance decomposition, we discovered that mode collapse does not appear in these cases. However, the lack of training data is still expressed in an increased bias term. Consequently, we are still convinced that our decomposition is a useful tool to gain insights into the fitting behavior of generative models.

---

> ### Author Response · Authors · 2023-11-17
> **Uncertainty Estimation in Image, Audio, and Natural Language Experiments**
>
> We now clarify how the experiments for image and audio are connected to the natural language generation (nlg) ones in terms of uncertainty estimation. In general, a meaningful measure of uncertainty is able to predict the loss for a given prediction.
> The literature formulates nlg uncertainty estimation as a binary classification problem with a binary loss (Kadavath et al, 2022; Kuhn et al, 2023), i.e. a generation is either correct or wrong. Consequently, it makes sense to use the AUROC as a summary measure of how well an uncertainty measure predicts the correctness (i.e. is correlated to the loss). For image and audio generation we use a kernel-based loss function (kernel score), which has a continuous range of how good a prediction is. Consequently, we cannot use the AUROC anymore. Instead, we use the Pearson correlation (range between -1 and 1) to assess how well an uncertainty measure can predict the loss. Absolute values close to 1 indicate a very good uncertainty estimate.
> In Figure 4 and 6, we can see that kernel entropy is very predictive of the loss (absolut Pearson correlation >0.9), making it an excellent measure of uncertainty in these cases. The nlg experiments in Figure 7 confirm that kernel entropy also gives state-of-the-art results for the binary loss function proposed by Kuhn et al (2023).
>
> **References:**
>
> Saurav Kadavath, Tom Conerly, Amanda Askell, Tom Henighan, Dawn Drain, Ethan Perez, Nicholas Schiefer, Zac Hatfield-Dodds, Nova DasSarma, Eli Tran-Johnson, et al. Language models (mostly) know what they know. arXiv preprint arXiv:2207.05221, 2022.
>
> Lorenz Kuhn, Yarin Gal, and Sebastian Farquhar. Semantic uncertainty: Linguistic invariances for uncertainty estimation in natural language generation. In The Eleventh International Conference on Learning Representations, 2023.

---

### Author Response · Authors · 2023-11-17
**Extended Experimental Details (Reviewers XNge, 27VK)**

We agree that the experimental details are not sufficient to keep our work self-contained.
To free up space, we move the bias-variance decomposition of the reproducing kernel Hilbert space (Paragraph around Equation 7) to Appendix B since it offers little additional insights for the practitioner compared to Theorem 3.2.
We use the gained space to provide more details on the experiments, including a major extension of the natural language experiment details in Section 5.3:

“An instance-level uncertainty measure is supposed to predict the correctness of an individual prediction and should therefore be highly correlated to the loss.
In all experiments so far, we observed a very high correlation between kernel entropy and kernel score.
This indicates that kernel entropy is an excellent measure of uncertainty.
In the following, we examine kernel entropy to predict the correctness of LLMs on question answering datasets.
Here, the setup differs from the previous experiments by two aspects.
First, we follow Kuhn et al. (2023) and do not use a kernel score as loss but a binarized version of the $\operatorname{RougeL}$ loss.
For two sequences $s, t$, it is defined as $\operatorname{RougeL} \left(s, t \right) = \frac{2}{\operatorname{length} \left( s \right) + \operatorname{length} \left( t \right)} \operatorname{LCS} \left(s, t \right)$, where $\operatorname{LCS}$ is the length of the **l**ongest **c**ommon **s**equence between its two inputs (Lin et al, 2004).
Kuhn et al (2023) propose to use the binary loss $L ( answer, target ) = \mathbf{1}_{ \operatorname{RougeL} \left(answer, target \right) > 0.3 }$.
This turns predicting the loss value into a binary classification problem.
Consequently, the Pearson correlation is not appropriate anymore to assess uncertainty measures.
Instead, the AUROC is used to evaluate the performance of uncertainty measures (Kadavath et al, 2022; Kuhn et al, 2023).”

**References:**

Lorenz Kuhn, Yarin Gal, and Sebastian Farquhar. Semantic uncertainty: Linguistic invariances for uncertainty estimation in natural language generation. In The Eleventh International Conference on Learning Representations, 2023.

Chin-Yew Lin and Franz Josef Och. Automatic evaluation of machine translation quality using longest common subsequence and skip-bigram statistics. In Proceedings of the 42nd Annual Meeting of the Association for Computational Linguistics (ACL-04), pp. 605–612, 2004.

Saurav Kadavath, Tom Conerly, Amanda Askell, Tom Henighan, Dawn Drain, Ethan Perez, Nicholas Schiefer, Zac Hatfield-Dodds, Nova DasSarma, Eli Tran-Johnson, et al. Language models (mostly) know what they know. arXiv preprint arXiv:2207.05221, 2022.

---

### Author Response · Authors · 2023-11-17
**General Response**

We thank all the reviewers for their time and efforts in reviewing our paper. We appreciate the insightful feedback and constructive suggestions. We are encouraged by your positive feedback on our work, which is “well-motivated” (XNge, vzDu) and “clearly written” (all reviewers).
We are pleased that the reviewers appreciate our introduced statistical tool as “useful” (vzDu, 27VK), which “compares favourably” (XNge, 27VK) with state-of-the-art approaches.

We remain convinced that our approach is of strong interest to the ICLR community, since our topic is “relevant” (XNge), “crucial” (ox7H), and “worth studying” (vzDu). We will first address the shared concern as part of the general response in *Extended experimental details*.
We then provide separate responses to each reviewer's individual questions. We also **updated the paper** and extended the appendix to address the raised concerns (**all changes marked in red**).

---

### Meta-Review · Area_Chair_XMpF · 2023-12-05

**Metareview:**

The authors use 'kernel entropy' (which is a kind of l2 variance estimate) as an uncertainty measure for outputs from a generative model like diffusions and language models. The authors findings are mode collapse of minority groups in diffusions, and better uncertainty quantification in triviaQA and coQA.

The strengths are the important problem setting, and outperforming existing uncertainty methods (semantic entropy, etc) on standard evaluations.

The reviewers note the somewhat scattered experiment design (the only 'end to end' performance evaluations are in the language setting, diffusion experiments done in one model, etc)

**Justification For Why Not Higher Score:**

The paper might have a good idea, but the experiment design and claims are overall not very polished. The authors fixed up some things during the rebuttal (tightening up the claims for the diffusion part and such) but this paper should probably go through another round of careful revisions and broadening the experiments.

**Justification For Why Not Lower Score:**

N/A

---

### Decision · Program_Chairs · 2024-01-16

Reject